## OPEN
# The globalizability of temporal discounting

**Economic inequality is associated with preferences for smaller, immediate gains over larger, delayed ones. Such temporal discounting may feed into rising global inequality, yet it is unclear whether it is a function of choice preferences or norms, or rather the absence of sufficient resources for immediate needs. It is also not clear whether these reflect true differences in choice patterns between income groups. We tested temporal discounting and five intertemporal choice anomalies using local currencies and value standards in 61 countries ($N = 13,629$). Across a diverse sample, we found consistent, robust rates of choice anomalies. Lower-income groups were not significantly different, but economic inequality and broader financial circumstances were clearly correlated with population choice patterns.**

Effective financial choices over time are essential for securing financial well-being[1,2], yet individuals often prefer immediate gains at the expense of future outcomes[3,4]. This tendency, known as temporal discounting[5], is often treated as a behavioural anomaly measured by presenting a series of choices that vary values, timelines, framing (for example, gains or losses) and other trade-offs[6]. Responses can then be aggregated or indexed in ways that test different manifestations of the anomaly, whether strictly the trade-off of immediate versus future or the threshold at which individuals are willing to change their preference[6].

Anomalies identified under temporal discounting are routinely associated with lower wealth[7–14], which is especially concerning given incongruent impacts on economic inequality brought about by the COVID-19 pandemic[15]. Inequality and low incomes have also routinely been associated with greater discounting of future outcomes[13,16,17], so it is not surprising that global studies would find temporal discounting (to varying degrees) in populations around the world[8]. However, the prevailing interpretations (that is, that lower-income groups show more extreme discounting[18,19]) may result from narrow measurement approaches, such as only assessing immediate gains versus future gains.

Another limitation of interpretations regarding discounting and economic classes involves the relative aspect of financial choices compared to income and wealth. Consider the patterns presented in Fig. 1a, which represent six months of spending patterns for 15,568 individuals in the United States who received stimulus payments as part of the 2020 CARES Act[20]. If the average amount spent 60 days prior to receiving the payment is used as a baseline, the lower-income group spent over 23 times more than baseline immediately after receipt, compared with around 10 times more than baseline for middle- and higher-income individuals. Apart from those days immediately following receipt, the relative spending patterns are almost identical for all three groups. However, as indicated on the right, those with higher incomes spent more in raw values, indicating that behaviours are more extreme only relative to income, and in fact, high-income individuals spent the most on average after receiving stimulus payments. While relative values may differentiate the consequences of spending, the spending patterns were generally about the same.

In this research, we aimed to test how broadly generalizable patterns of temporal discounting are around the world, incorporating social and economic factors as well as multiple measures of intertemporal choice. With broader testing of more anomalies, rather than being limited to indifference points (a threshold value for preferring now versus later), more robust conclusions can be drawn about choice patterns. In this vein, the most comprehensive related study found that lower-income countries had lower trust in systems and had the steepest rates of discounting (that is, the threshold for giving up an immediate gain for a later, larger one was much higher)[8,21]. As the indifference point was the primary indicator, these results are extremely important but do not necessarily mean that lower-income populations have distinct decision-making patterns. Three similar studies also tested temporal choice in large, multi-national populations, some including more than 50,000 participants from more than 50 countries[18,22]. These studies largely focused on smaller-sooner versus larger-later constructs of temporal discounting. Most concluded that lower income and wealth, among other micro and macro variables, were strong predictors of higher discounting (or lower patience). However, these studies did not incorporate a broad range of temporal choice constructs, as their focus was typically specific to time preferences.

To avoid the limitations of relying only on indifference points and to assess the generalizability of temporal discounting on a near-global scale, we used a similar method to those studies but tested multiple intertemporal choice domains. Our approach allows the rates of certain anomalies to be considered along with specific value thresholds. Our aim was to test each of these patterns for generalizability while also factoring in multiple economic aspects across populations, primarily wealth, inequality, debt and inflation. We pre-registered (https://osf.io/jfvh4) six primary hypotheses, anticipating that temporal discounting would be observed in all countries to varying extents, though mean differences between countries would be less extreme than variability within countries, both overall and for specific anomalies. We also anticipated that economic inequality would be a strong predictor of national discounting averages.

Inflation, which tends to be higher in lower-income countries[23], is also associated with stronger preferences for immediate gains[24,25]. In our final hypothesis, we expected to confirm this pattern, indicating that such preferences may be associated with increased probability that future gains will be worth substantially less than their current value. We expected that this might be even more broadly impactful than income or wealth, though each interacts in some way and all should be considered. We limited our hypotheses to inflation versus extreme inflation: we expected that differences in preferences would emerge only at substantially larger inflation rates (over 10%) and hyperinflation (over 50%), and less so between regions with varied but less extreme differences (substantively below 10%).

To test our hypotheses, we used four choice anomalies outlined in one of the most influential articles[26] on intertemporal choice—absolute magnitude, gain–loss asymmetry, delay–speedup asymmetry and common difference (we refer to this as present bias, which

is the more common term)—plus a fifth, subadditivity, to complete three inter-related time intervals[27]. In contrast to most discounting research, using a series of intertemporal choice anomalies[28] identified in WEIRD labs allows us to test including patterns that choice models often ignore. When multiple anomalies are tested alongside a simplified indifference measure (derived from the first set of choices), the prevalence of each anomaly provides a more robust determination of the generalizability of the construct than an indifference point alone.

By addressing both the depth of the method used and concerns about the generalizability of behavioural research[29], the richer perspective of our approach to measuring intertemporal decision-making in a global sample allows us to assess the presence and prevalence of anomalies in local contexts. It also allows us to test potential relationships with economic inequality to determine whether low-income groups are somehow more extreme decision-makers or whether the environment, beyond simply individual circumstances, is a more impactful factor across populations.

Most research on temporal preferences uses indifference points[6], which determine the threshold at which individuals will shift from immediate to delayed (and vice versa). Data from that approach are robust and converge on an inverse relationship between income/wealth and discounting rate. However, multiple binary choice comparisons are ideal for demonstrating multidimensional choice patterns, as in prospect theory, expected utility and other choice paradoxes or cognitive biases. They are also better suited for testing in multiple countries[30,31] when multiple small adaptations to values in different currencies are necessary. Taking this into consideration, our method leveraged one of the most widely cited papers on decision-making[26], which proposed four critical intertemporal choice anomalies. While studies of individual anomalies exist from various regions[32–34], our approach aimed to produce a comprehensive multi-country assessment that simultaneously tested the generalizability of all four:

- Absolute magnitude: Increased preference for delayed gains when values become substantially larger, even when relative differences are constant (for example, prefer $500 now over $550 in 12 months and prefer $5,500 in 12 months over $5,000 now[4,7]).
- Gain–loss asymmetry: Gains are discounted more than losses, though differences (real and relative) are constant (for example, prefer to receive $500 now over $550 in 12 months, but also prefer to pay $500 now over paying $550 in 12 months).
- Delay–speedup asymmetry: Accepting an immediate, smaller gain if the delay is framed as added value, but preferring the larger, later amount if an immediate gain is framed as a reduction (for example, prefer to receive a gain of $500 rather than wait 12 months for an additional $50 and prefer to wait for 12 months to receive $550 rather than to pay $50 and receive the gain now).
- Present bias: Lower discounting over a given time interval when the start of the interval is shifted to the future (for example, prefer $500 now over $550 in 12 months and prefer $550 in two years over $500 in 12 months).

We also assess subadditivity[27] effects, which adds an interval of immediate to 24 months, thereby allowing us to fully assess discounting over three time intervals (0–12, 12–24 and 0–24 months)[35]. Subadditivity is considered present if discounting is higher for the two 12-month intervals than for the 24-month interval.

All data were collected independent of any other study or source, with a 30-item instrument developed specifically for assessing a base discounting level and then the five anomalies. To validate the metric, a three-country pilot study (Australia, Canada and the United States) was conducted to confirm that the method elicited variability in choice preferences. We did not assess what specific patterns of potential anomalies emerged to avoid biasing methods or decisions related to currency adaptations.

For the full study, all participants began with choosing either approximately 10% of the national monthly household income average (either median or mean, depending on the local standard) immediately, or 110% of that value in 12 months. For US participants, this translated into US$500 immediately or US$550 in one year. Participants who chose the immediate option were shown the same option set, but the delayed value was now 120% (US$600). If they continued to prefer the immediate option, a final option offered 150% (US$750) as the delayed reward. If participants chose the delayed option initially, subsequent choices were 102% (US$510) and 101% (US$505). This progression was then inverted for losses, with the same values presented as payments, increasing for choosing delayed and decreasing for choosing immediate. Finally, the original gain set was repeated using 100% of the average monthly income to represent higher-magnitude choices (Supplementary Table 1).

After the baseline scenarios, the anomaly scenarios incorporated the simplified indifference point (the largest value at which the participants chose the delayed option in the baseline items; see Supplementary Methods). Finally, the participants answered ten questions on financial circumstances, (simplified) risk preference, economic outlook and demographics. The participants could choose between the local official language (or languages) and English. By completion, 61 countries (representing approximately 76% of the world population) had participated (Supplementary Tables 2 and 3).

We assessed temporal choice patterns in three ways. First, we used the three baseline scenarios to determine preferences for immediate or delayed gains (at two magnitudes) and losses (one). Second, we calculated the proportion of participants who exhibited the theoretically described anomaly for each anomaly scenario (Supplementary Table 4). We also calculated proportions of participants who exhibited inconsistent decisions even if not specifically aligned with one of the defined anomalies. Finally, we computed a discounting score based on responses to all choice items, ranging from 0 (always prefer delayed gains or earlier losses) to 19 (always prefer immediate gains or delayed losses). The score then represents the consistency of discounting behaviours, irrespective of the presence of other choice anomalies (see Supplementary Information for details on reliability and validity).

To explore individual and country-level differences, we performed a series of multilevel linear and generalized mixed models that predicted standardized temporal discounting scores and anomalies, respectively. We ran a set of increasingly complex models, including inequality indicators, while controlling for individual debt and assets, age, education, employment, log per-capita gross domestic product (GDP) and inflation at the individual and country levels. Because the raw scores (0–19) have no standard to compare against, we primarily used standardized scores (with a mean of 0 and standard deviation of 1) for analysis and visualization.

We detected several relevant nonlinear effects (debt, financial assets and inflation; Supplementary Tables 5–7), which we incorporated into our final models via spline modelling[36]. The models were estimated using both frequentist (Supplementary Tables 8 and 9 and Supplementary Figs. 1 and 2) and Bayesian techniques (Supplementary Tables 10 and 11), assessing the consistency of the results. Support for potential null effects was evaluated using a variety of Bayesian approaches (Supplementary Table 12).

There are some limitations in our approach. The most noteworthy is that we are limited to hypothetical scenarios in which the participants had no motivation to give a particular answer, which might have impacted responses had true monetary awards been offered. Though Japanese participants received payment, it was not contingent on their choices, so the same limitation holds. While that might have been an ideal approach, substantial evidence indicates that such hypothetical scenarios do not differ substantively from

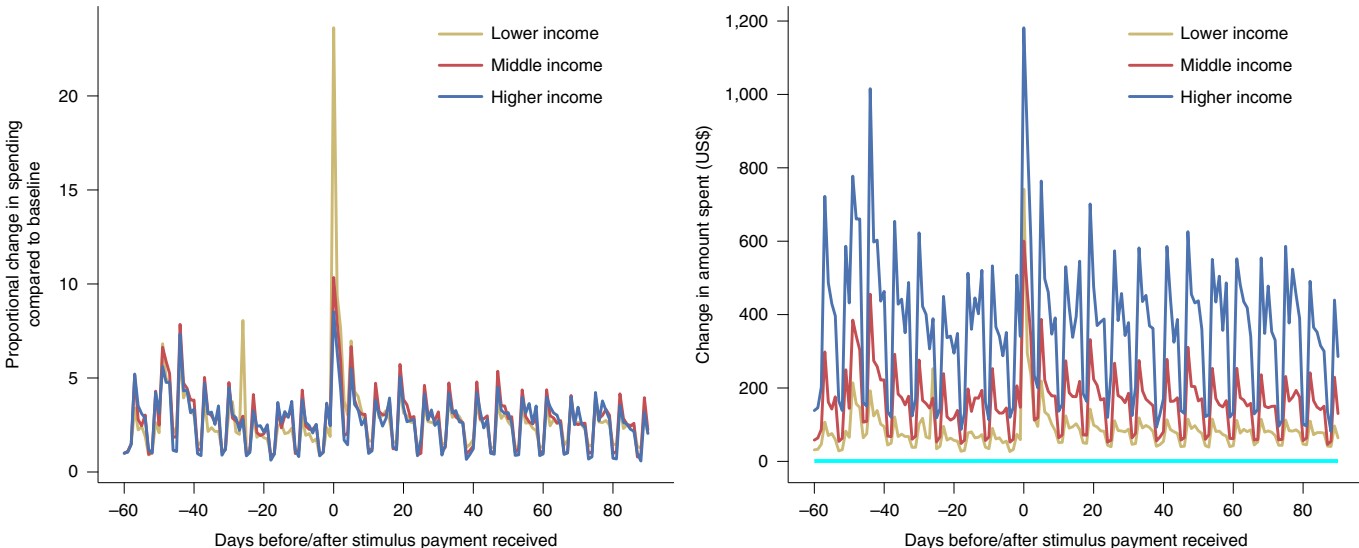

**Fig. 1 | Spending timelines after receiving the COVID-19 relief stimulus payment.** Spending before and after receiving a 2020 CARES Act stimulus payment for lower-income (earning under US$28,001 per year), middle-income (US$28,001–US$68,000) and higher-income (above US$68,000) individuals. The baseline average (light blue line) is the amount spent 60 days prior to receiving the payment. The left plot presents proportional spending compared with a standard baseline. The right plot presents the same information but uses actual spending values. Apart from the days immediately following receipt, the base-standardized spending patterns are almost identical for all three groups.

actual choices, and many such approaches have been validated to correlate with real-world behaviours[37–42]. Naturally, this does not provide a perfect replacement for comprehensive real-world behavioural observations, but there is sufficient evidence to indicate that hypothetical approaches yield reasonably valid results. The second limitation is that our approach to minimizing bias through highly randomized and broad data collection yielded demographics that varied in representativeness. For indications of how this may have impacted the results, we included a complementary demographics table for comparison between the sample and true national characteristics (Supplementary Table 18).

Finally, in terms of robustness in our methods, we opted for five anomalies tested in relatively short form rather than a smaller number of domains in long form. We did this in part because it would be a meaningful contribution to the field as well as because it was more important to demonstrate the existence of anomalies than to emphasize precise thresholds (for example, indifference points). Though it was impractical to do comprehensive, adaptive measures for our approach, we strongly encourage future studies involving both a broad number of choice domains and extensive measures within each to offer greater precision.

## Results

For 13,629 participants from 61 countries, we find that temporal discounting is widely present in every location, indicating consistency and robustness (with some variability) across all five intertemporal choice anomalies (Fig. 2). Income, economic inequality, financial wealth and inflation demonstrated clear links to the shape and magnitude of intertemporal choice patterns. Better financial environments were consistently associated with lower rates of temporal discounting, whereas higher levels of inequality and inflation were associated with higher rates of discounting. Yet, the overall likelihood of exhibiting anomalies remained stable irrespective of most factors.

Differences between locations are evident, though remarkable consistency of variability exists within countries. Such patterns demonstrate that temporal discounting and intertemporal choice anomalies are widely generalizable, and that differences between

individuals are wider than differences between countries. Being low-income is not alone in relating to unstable decision-making; being in a more challenging environment is also highly influential.

The scientific and policy implications from these findings challenge simple assumptions that low-income individuals are fundamentally extreme decision-makers. Instead, these data indicate that anyone facing a negative financial environment—even with a better income within that environment—is likely to make decisions that prioritize immediate clarity over future uncertainty. While we do not explicitly test risk in the temporal measures, all future prospects inherently hold a risk component, which is compounded by temporal distance and environmental instability (that is, the further the distance between two prospects and the less stable the future may be, the greater the inherent risk difference may be perceived between an immediate and a future prospect)[43–45]. Likewise, the data indicate that all individuals at all income levels in all regions are more likely than not to demonstrate one or more choice anomalies.

**Detailed analysis of temporal choice anomalies.** We collected 13,629 responses from 61 countries (median sample size of 209, Supplementary Tables 2 and 3). Though the absolute minimum sample size necessary was 30 per country, the sliding scale used for ensuring full power (see Selection of countries) started at 120, increasing to 360 for larger countries. Forty-six countries achieved the target sample size, and 56 had at least 120 (with at least four countries per continent at 120), thus providing a wide range of economic and cultural environments. Only two countries, where data collection was exceptionally challenging, had below 90 participants, but all locations were still substantially above the absolute minimum. As well as exceeding the minimum sample size, we chose to retain these participants in the analyses because they represent groups often not included in behavioural science[46,47].

In line with related research[8], Fig. 3 shows how countries with lower incomes typically had greater temporal discounting levels in the baseline items (Supplementary Table 14). This was most evident in the tendency to prefer immediate gains, even as delayed prospects increased. This pattern was not found for the loss scenario. However, as noted, these items give a useful measure for the

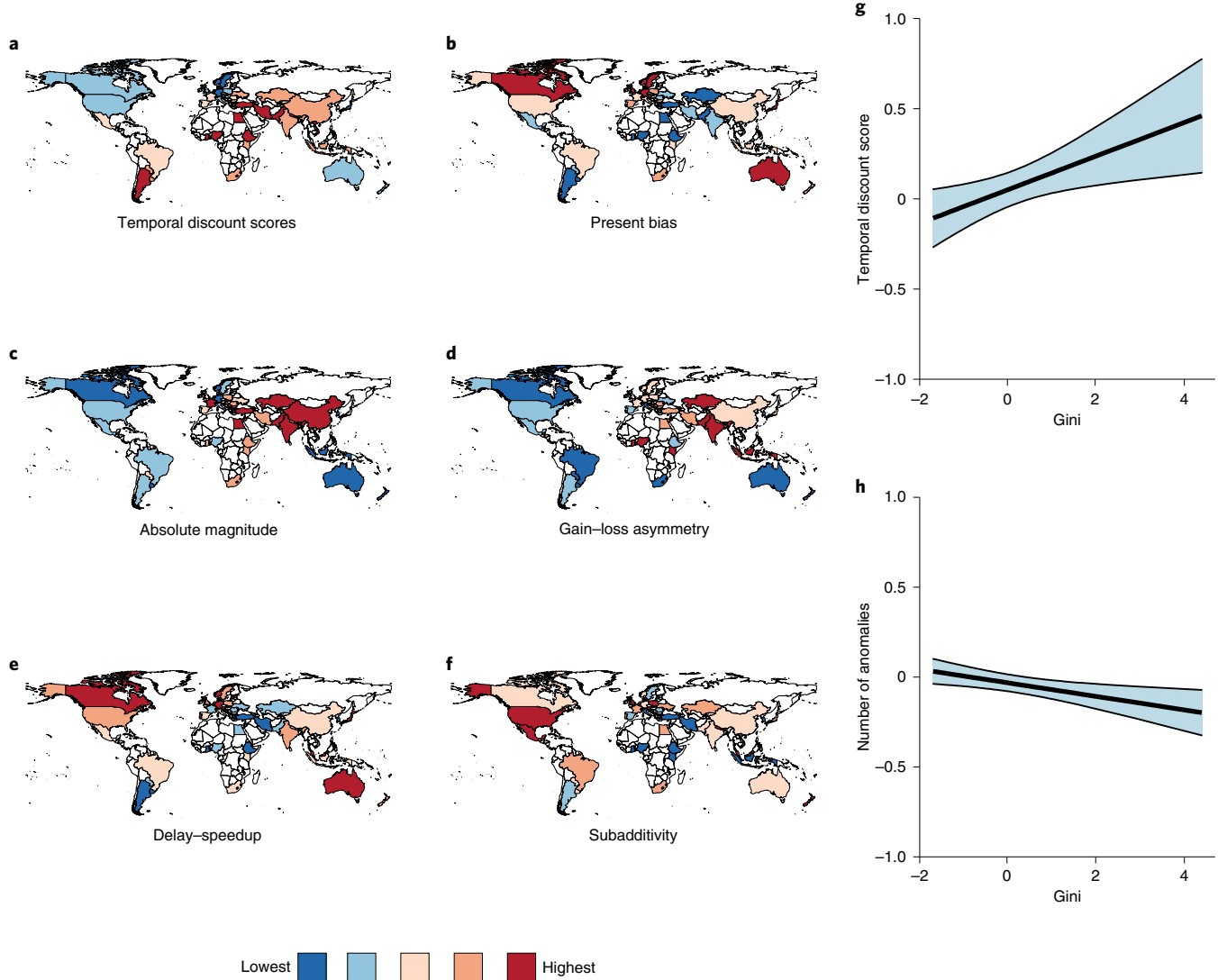

**Fig. 2 | Global indications of intertemporal choice. a–f**, Maps of choice preferences in aggregate and by individual anomaly indicate heterogeneity in intertemporal choice patterns. While some subtle patterns emerge, particularly stronger preferences for delayed gains in higher-income regions, choice preferences are broadly consistent across 61 countries in the sense that all anomalies appear in all locations. No location consistently presents extremes (high or low) of each anomaly. The results are based on the models specified in Supplementary Table 13. **g,h**, Conditional smooth effects (black) and 95% confidence intervals (light blue). Map from Natural Earth (naturalearthdata.com).

indifference level for each individual but do not give a robust indication of whether temporal choice anomalies are present.

Between-countries random-effect meta-analyses estimated pooled and unpooled effects for aggregate scores and individual anomalies (Supplementary Figs. 3–8). Temporal discounting was present in all countries, with only modest variability in national means (aggregate mean, 10.3; prediction interval, (6.8, 13.8); from Japan (mean = 7.1, s.d. = 3.9) to Argentina (mean = 14.1, s.d. = 3.0); Fig. 4). Overall, 54% of participants showed at least one anomaly, with 33% presenting multiple and only 2% showing four (Supplementary Table 15). Anomalies were present in all locations, and aggregate values indicated the widespread presence of the four primary anomalies (from 13.8% for absolute magnitude to 40.1% for gain–loss asymmetry, Fig. 3). Gain–loss rates were the most common anomaly in 80.3% (49) of the countries, with substantially higher rates observed than for the other anomalies. While only 10.7% of the sample engaged in subadditivity behaviour (range, 2.7% (Lebanon) to 20.7% (New Zealand)), the criteria were stricter for this anomaly.

In all cases, significant $Q$-tests and $I^2$ values over 70% suggested that effect size variation at the country level could not be accounted for by sampling variation alone. There were strong relationships between the individual and aggregate scores and some anomalies (that is, positive for absolute magnitude and negative for present bias and delay–speedup; Supplementary Fig. 9). Additionally, we found a negative link between GDP and temporal discount scores ($\beta = -0.07$; $P = 0.001$; 95% confidence interval, (−0.12, −0.03)), and positive effects for present bias (odds ratio (OR), 1.09; $P = 0.003$; 95% confidence interval, (1.03, 1.16)) and delay–speedup (OR = 0.95; $P = 0.002$; 95% confidence interval, (0.91, 0.99)). We found no evidence of an association for the remaining anomalies (0.95 < OR < 1.01, 0.027 < $P$ < 0.688). We note that some ORs in the non-significant anomalies were similar to those that were significant, but given the sample size, we adhered to a strict cut-off for significance; future research may benefit from reanalysing these data within each country to explore whether more delineated patterns may exist between aggregate wealth and temporal choice anomalies.

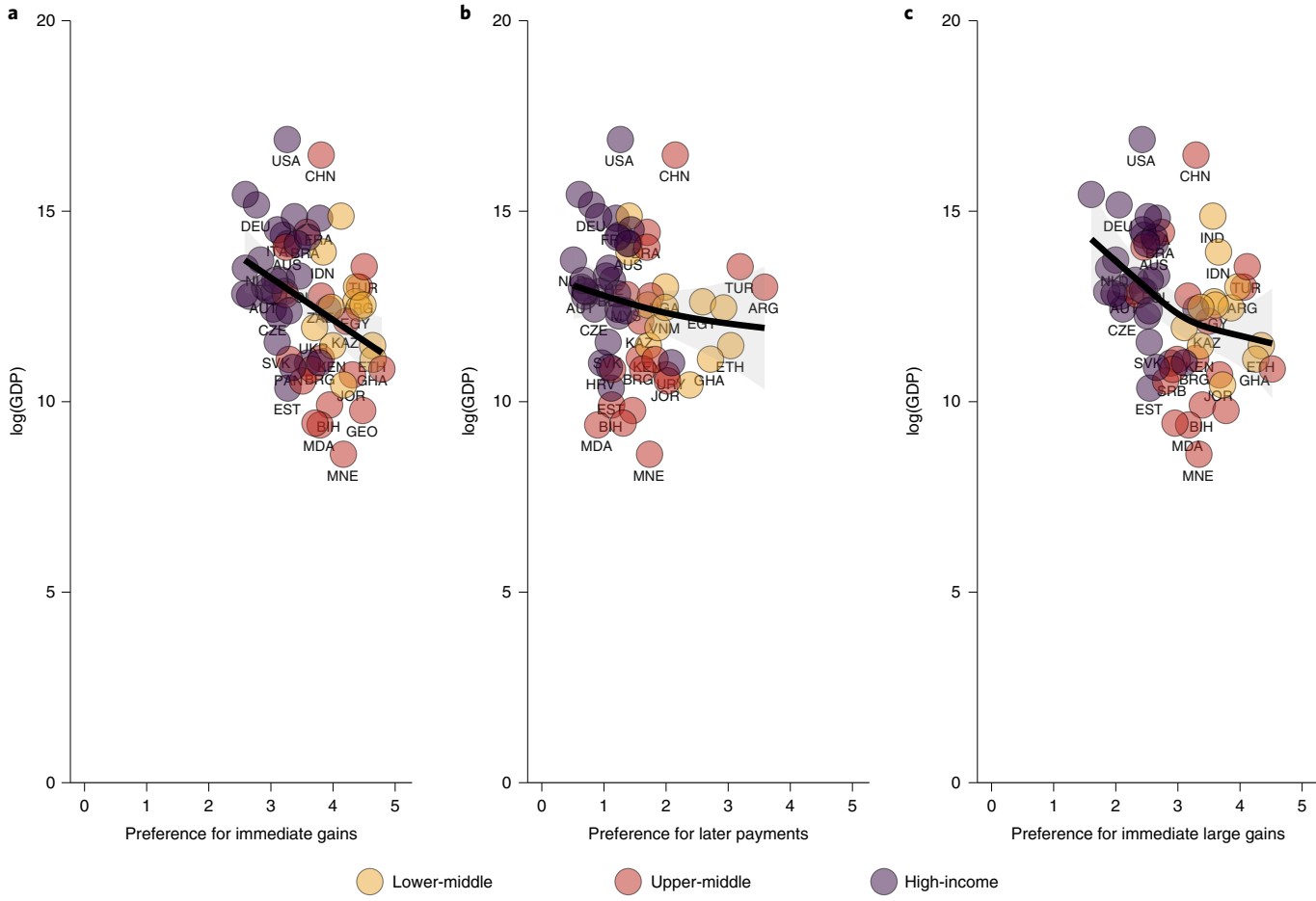

**Fig. 3 | Baseline temporal discounting and GDP. a–c**, There is a clear trend of lower GDP[36] being associated with higher preferences for immediate gains and later payments. However, all locations indicate some preference for immediate over delayed. Taken together, this provides support for the hypothesis that baseline temporal discounting is observed globally and that the economic environment may shape its contours. The results are based on the models specified in Supplementary Table 14. Smooth terms and 95% confidence intervals are presented in black and grey, respectively.

Despite between-country differences in mean scores and anomaly rates, there was substantial overlap between response distributions. Accordingly, results from multilevel models indicated that no more than 20% of the variance was ever explained by between-country differences for scores and was between 2% (absolute magnitude) and 8% (present bias) for anomalies. We thus find temporal discounting to be globally generalizable, robust and highly consistent (in line with expectations) (Supplementary Table 6 and Supplementary Fig. 10), where within-country differences between individuals are substantially greater than between-country differences. In other words, we find temporal discounting to be a globalizable (though not universal) construct. We also find that there is nothing WEIRD about intertemporal choice anomalies.

*Inequality.* We defined inequality at the level of the country and at the level of the individual. For countries, we used the most recently published Gini coefficients[48]. For individuals, we calculated the difference between their reported income and the adjusted net median local (country) income. At the country level, Gini had a positive relationship with temporal discounting scores ($\beta = 0.09$; $P = 0.002$; 95% confidence interval, (0.02, 0.06); Supplementary Table 8), yet no such pattern emerged for specific anomalies, as we observed no significant effect for the remaining cases ($0.92 < OR < 1.01$, $0.023 < P < 0.825$, Supplementary Table 8). Individual income inequality did not predict temporal discounting scores ($\beta = -0.01$;

$P = 0.121$; 95% confidence interval, ($-0.03$, 0.001)) or rates of anomalies ($0.96 < OR < 1.04$, $0.045 < P < 0.867$, Supplementary Tables 8 and 9), except two small effects for present bias ($OR = 1.07$; $P = 0.006$; 95% confidence interval, (1.03, 1.13)) and absolute magnitude ($OR = 0.92$; $P = 0.006$; 95% confidence interval, (0.87, 0.98); Supplementary Table 9).

As shown in Fig. 5, these patterns are largely in line with expectations, indicating that, in aggregate, greater inequality is associated with increased rates of discounting. However, as indicated in Fig. 3, intertemporal choice anomalies overall are not unique to a specific income level, and worse financial circumstances may be associated with more consistent choice patterns (that is, fewer anomalies) due to sustained preference for sooner gains. Whether this aligns with arguments that scarcity leads individuals to focus on present challenges is worthy of further exploration[49]. It also reiterates that patterns in population (that is, country) aggregates are not the same as predicting individual choices[50].

*Assets and debt.* We found consistently that greater willingness to delay larger gains tends to be associated with greater wealth (financial assets), except for the extremely wealthy. Temporal discounting scores generally decreased as wealth increased, except for the wealthiest individuals (expected degrees of freedom (e.d.f.) (see 'Further details on modeling temporal discounting' in the Supplementary Information), 2.88; $P < 0.0001$; Supplementary Table 8 and

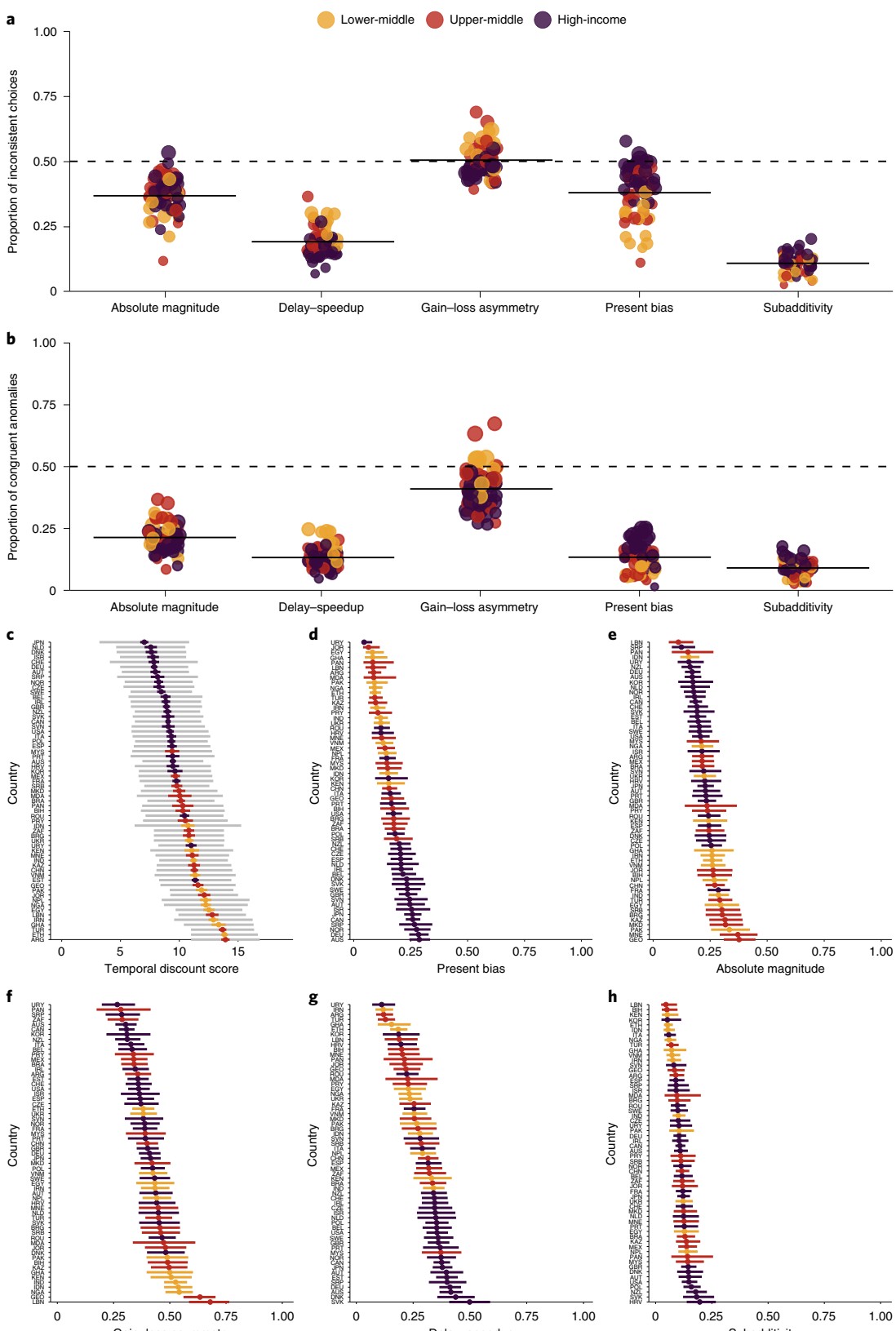

**Fig. 4 | Anomalies and temporal discounting scores by country. a,b,** Proportions (solid bars are overall means) of participants that demonstrated inconsistent choice preferences (**a**) and the proportion of each country sample that aligned with the five anomalies of interest (**b**). Apart from absolute magnitude and present bias, no consistent rate was based on wealth, and all countries indicate some presence of each anomaly. **c–h,** Each plot presents the distribution of values ordered by mean or proportion value. Plot **c** presents the distribution of discounting scores for each country, including means, prediction intervals (coloured) and standard deviations (grey). Plots **d–h** show the proportions of participants that presented each anomaly. While the difference from lowest to highest for each is noteworthy, similar variabilities exist across all. See Supplementary Figs. 3–8 for the full values and sample sizes for each point.

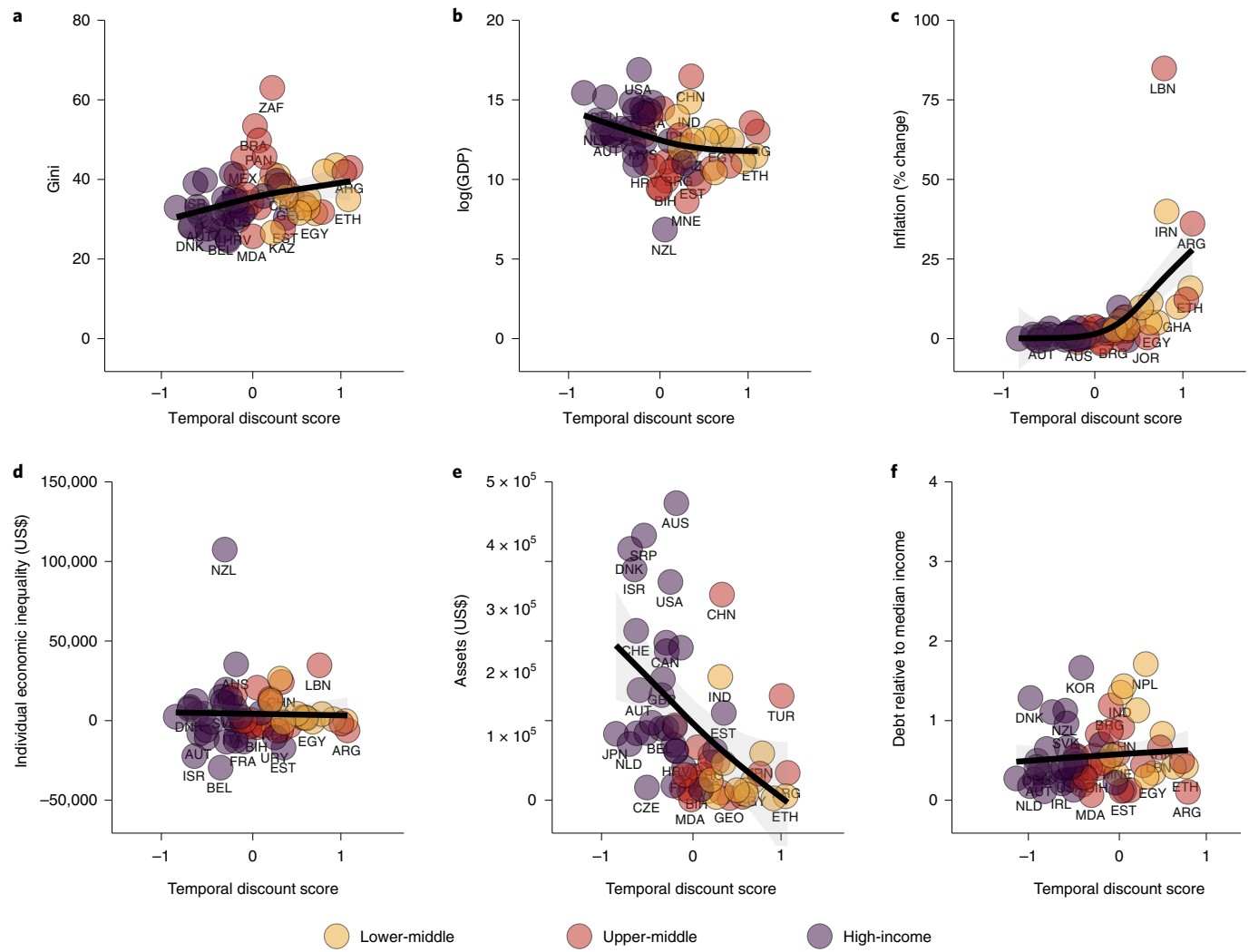

**Fig. 5 | Wealth, debt, inequality and temporal discounting. a–f**, Plots using standardized scores for temporal discounting indicate an overall trend that greater wealth and income at the individual and national levels are associated with lower overall temporal discounting, and greater economic inequality and individual debt are associated with lower overall temporal discounting. Inflation has a modest relationship with discounting, which becomes much stronger at substantially high levels of inflation. The results for each variable by score are from models specified in Supplementary Table 16. Smooth terms and 95% confidence intervals are presented in black and grey, respectively.

Supplementary Fig. 2). We also observed assets being associated with present bias (e.d.f. = 1.01, $P < 0.0001$) and with delay–speedup (e.d.f. = 2.78, $P < .0001$). We observed the reverse pattern for absolute magnitude (e.d.f. = 1.96, $P = 0.0009$). For gain–loss asymmetry (e.d.f. = 0.474, $P = 0.144$) and subadditivity (e.d.f. = 0.001, $P = 0.472$), we found no meaningful relationship between assets and the likelihood of observing either (Supplementary Table 9 and Supplementary Fig. 2). Higher levels of debt were associated with lower discount rates, particularly for people with lower to medium debt (e.d.f. = 2.91, $P < 0.0001$, Supplementary Fig. 1), though there was no significant effect observed regarding debt and the likelihood of engaging in any specific anomaly ($0.95 < \mathrm{OR} < 1.01$, $0.035 < P < 0.944$, Supplementary Table 9).

*Inflation.* We observed strong relationships between inflation rates and temporal discounting scores as well as all anomalies. There was a particularly strong effect of hyperinflation on temporal discounting (e.d.f. = 1.81, $P < 0.0001$, Supplementary Table 8 and Supplementary Fig. 1), with some levelling out at the extremes. Countries experiencing severe hyperinflation demonstrate extreme

discounts only for gains but not for payments, which minimizes the effect on total scores. However, if limiting to only gains, the effect remains extreme, as indicated by the two gain scenarios in Fig. 3.

We observed a reverse trend of higher inflation being associated with a lower likelihood of engaging in anomalies (Supplementary Table 9 and Supplementary Fig. 2)—namely, for present bias (e.d.f. = 1.63, $P < 0.0001$), absolute magnitude (e.d.f. = 1.92, $P < 0.0001$), delay–speedup (e.d.f. = 1.75, $P < 0.0001$) and subadditivity (e.d.f. = 1.37, $P = 0.0019$). The only positive (but weaker) effect in the case of anomalies was found for gain–loss asymmetry (e.d.f. = 1.675, $P = 0.0051$).

## Discussion
For good reason, psychological theory has come under considerable recent criticism due to a number of failed replications of previously canonical constructs[51]. There is also wide support to consider that the absence of testing (or adapting methods to test) across populations limits the presumed generalizability of conclusions in the field[29]. To the extent that it is possible for any behavioural phenomenon, we find temporal discounting and common intertemporal

choice anomalies to be globally generalizable. This is largely based on finding remarkable consistency and robustness in patterns of intertemporal choice across 61 countries, with substantially more variability within each country than between their means. We emphasize that while discounting may be stronger in worse financial circumstances, particularly those with poorer economic outlooks, it exists in all locations at measurable levels.

We do not imply that temporal discounting and specific intertemporal choice anomalies are universal (that is, present in all individuals at all times). Instead, our findings provide extreme confidence that the constructs tested are robust on a global level. In our view, they also disrupt some notions that lower-income individuals are somehow inherently unstable decision-makers, as negative environments are widely influential. Under such circumstances, it is both rational and, as our data show, entirely typical to follow the choice preferences we present.

We hope these findings will be considered in both science and policy, particularly in how governments and institutions can directly impact inequality. Consider excessive savings requirements to acquire mortgages[52], less favourable lending terms for low earners[53], harmful interest rates on financing necessities such as education, restricting access to foreign currency and focusing taxes on income without considering wealth, assets or capital[54]. Some of these are based on assumptions of how income and wealth are primary indicators of long-term decision-making, but in fact those policies alone can create economic barriers that impact upward economic mobility. On top of impeding mobility, these policies risk institutional resilience by offering better terms (and therefore taking on greater risk) to higher-wealth groups on the basis of reductionist presumptions about who has the lowest discounting rates, or ignoring how inflation may impact spending and saving behaviours among the most financially vulnerable.

The scope of the work, particularly the diversity of these 13,629 participants across 61 countries, should encourage more tests of global generalizability of fundamental psychological theory that adapt to local standards and norms. Similarly, policymakers should consider the effects of economic inequality and inflation beyond incomes and growth and give greater consideration to how they directly impact individual choices for entire populations, affecting long-term well-being.

## Methods

Ethical approval was given by the Institutional Review Board at Columbia University for both the pilot study and the full study. For the full study, all countries involved had to provide attestations of cultural and linguistic appropriateness for each version of the instrument. Because this was not possible for the pilot study, ethical approval was given only to check the quality, flow and appropriateness of the survey instrument, but not to analyse or report data. For all data, all participants provided informed consent at the start of the survey, and no forms of deception or hidden purpose existed, so all aspects were fully explained.

The materials and methods followed our pre-registered plan (https://osf. io/jfvh4). Substantive deviations from the original plan are highlighted in each corresponding section, alongside the justification for the deviation. All details on the countries included, translation, testing and sampling are included in the Supplementary Information.

**Participants.** The final dataset was composed of 13,629 responses from 61 countries. The original sample size was 25,877, which was reduced almost by half after we performed pre-registered data exclusions. We removed 6,141 participants (23.7%) who did not pass our attention check (a choice between receiving 10% of monthly income now or paying the same amount in one year). We removed 69 participants for presenting non-sensical responses to open data text (for example, 'helicopter' as gender). We removed 13 participants claiming to be over 100 years old. We included additional filters to our original exclusion criteria. Regarding the length of time for responses, individuals faster than three times the absolute deviation below the median time or that took less than 120 seconds to respond were removed. This criterion allowed us to identify 5,870 inappropriate responses. We further removed responses from IP addresses identified as either 'tests' or 'spam' by the Qualtrics service (264 answers identified). Lastly, we did not consider individuals not completing over 90% of the survey (9,434 responses failed this

criterion). Note that these values add up to more than 100% because participants could fail multiple criteria.

For analyses including income, assets and debt, we conducted additional quality checks. We first removed 38 extreme income, debt or assets (values larger than $1 \times 10^8$) responses. Next, we removed extreme outliers larger than 100 times the median absolute deviation above the country median for income and 1,000 times larger than the median absolute deviation for national median assets. We further removed anyone that simultaneously claimed no income while also being employed full-time. These quality checks identified 54 problematic responses, which were removed from the data. The final sample and target size are presented in Supplementary Table 2. We provide descriptive information on the full and by-country samples in Supplementary Table 3 and the main variables in Supplementary Table 4.

**Instrument.** The instrument was designed by evaluating methods used in similar research, particularly those with a multi-country focus[8,21,29] or that covered multiple dimensions of intertemporal choice[13,28]. On the basis of optimal response and participation in two recent studies[6,49] of a similar nature, we implemented an approach that could incorporate these features while remaining brief. This design increased the likelihood of reliable and complete responses.

To confirm the viability of our design, we assessed the overall variability of pilot study data from 360 participants from the United States, Australia and Canada. The responses showed that the items elicited reasonable answers, and the three sets of baseline measures yielded responses that would be expected for the three countries. Specifically, it was more popular to choose earlier gains over larger, later ones for the smaller magnitude and closer to 50–50 for the larger magnitude and the payment set. The subsequent choice anomalies also yielded variability within items, which showed some variability between countries. These results confirmed that using baseline choices to set trade-off values in anomaly items was appropriate and would capture relevant differences. We did not analyse these data in full per our Institutional Review Board approval, as we did not want a detailed analysis of subsequent bias decisions. The pilot was completed in April 2021 with participants on the Prolific platform (compensated for participation, not for choices made).

The final version of the instrument required the participants to respond to as few as 10 to as many as 13 anomaly items. All items were binary. During the first three anomaly sets, if a participant chose immediate and then delay (or vice versa), they proceeded to the next anomaly, so only two questions were required. If they decided on immediate–immediate or delay–delay, they would see the third set. After the anomalies, the participants answered ten questions about financial preferences, circumstances and outlook (most of these will be analysed in independent research). Finally, the participants provided age, race/ethnicity/immigration status, gender, education, employment and region of residence. Supplementary Table 1 presents all possible values for each set of items used in the final version of the instrument.

All materials associated with the method are available in the pre-registration repository.

**Selection of countries.** By design, there was no systematic approach to country inclusion. Through a network of early career researchers worldwide, multiple invitations were sent and posted to collaborate. We explicitly emphasized including countries that are not typically included in behavioural research, and in almost every location, we had at least one local collaborator engaged. All contributors are named authors.

Following data collection, 61 countries were fully included, using 40 languages. All countries also had an English version to include non-native speakers who were uncomfortable responding in the local language. Of the 61 countries, 11 were from Asia, 8 were from the Americas, 5 were from sub-Saharan Africa, 6 were from the Middle East and North Africa, 2 were from Oceania, and 29 were from Europe (19 from the European Union). Several additional countries were attempted but were unable to fulfil certain tasks or were removed for ethical concerns.

**Translation of survey items.** All instruments went through forward-and-back translation for all languages used. In each case, this required at least one native speaker involved in the process. All versions were also available in English, applying the local currencies and other aspects, such as race and education reporting standards. A third reviewer was brought in if discrepancies existed that could not be solved through simple discussion. Similar research methods were also used for wording. The relevant details where issues arose are included in the Supplementary Information. For cultural and ethical appropriateness, demographic measures varied heavily. For example, in some countries, tribal or religious categories are used as the standard. Other countries, such as the US, have federal guidelines for race and ethnicity, whereas France disallows measures for racial identity. The country-by-country details are posted on the pre-registration page associated with this project.

All data were collected through Qualtrics survey links. For all countries, an initial convenience sampling of five to ten participants was required to ensure that comprehension, instrument flow and data capture were functional. Minor issues were corrected before proceeding to 'open' collection. Countries aimed to

recruit approximately 30 participants before pausing to ensure functionality and that all questions were visible. We also checked that currency values had been appropriately set by inspecting responses' variability (that is, if options were poorly selected, this would be visible in having all participants make the same choices across items). Minimal issues arose and are outlined in the Supplementary Information.

For data circulation, all collaborators were allowed a small number of convenience participants. This decision limited bias while ensuring the readiness of measures and instruments, as multiple collaborators in each country used different networks, thereby reducing bias. Once assurances were in place, we implemented what we refer to as the Demić–Većkalov method, which two prior collaborators in recent studies developed. This method involves finding news articles online (on social media, popular forums, news websites, discussion threads, sports team supporter discussion groups/pages and so on) and posting in active discussions, encouraging anyone interested in the subject to participate. Circulation included direct contact with local organizations (non-governmental organizations and non-profits, often with thematic interests in financial literacy, microcredit and so on) to circulate with stakeholders and staff, email circulars, generic social media posts, informal snowballing and paid samples (in Japan only; no other participants were compensated). We note that this approach to data collection with a generally loose structure was intentional to avoid producing a common bias across countries. Similar to recent, successful multi-country trials[30,55], this generates more heterogeneous backgrounds, though it still skews toward populations with direct internet access (that is, younger, higher education and somewhat higher income).

As described in the pre-registration (https://osf.io/jfvh4), the minimum sample threshold to achieve a power of 0.95 for the models presented was 30 participants per country. However, to produce a more robust sample, we used three tiers for sample targets: population ≤ 10 million, 120 participants; 10 million ≤ population ≤ 100 million, 240 participants; and population > 100 million, 360 participants.

Comprehensive details about methods, guidelines, measurement building and instruments are available in the Supplementary Information and on the pre-registration site.

**Procedure.** For the full study, all participants began by choosing from two gains of approximately 10% of the national household income average (either median or mean, depending on the local standard) immediately, or 110% of that value in 12 months. For US participants, this translated into US$500 immediately or US$550 in one year. Participants who chose the immediate option were shown the same option set, but the delayed value was now 120% (US$600). If they preferred the immediate prospect, a final option offered 150% (US$750) as the delayed reward. If participants chose the delayed option initially, subsequent choices were 102% (US$510) and 101% (US$505). This progression was then inverted for losses, with the identical values presented as payments, increasing for choosing delayed and decreasing for choosing immediately. Finally, the original gain set was repeated using 100% of the monthly income to represent higher-magnitude choices.

Following the baseline scenarios, the anomaly scenarios incorporated the simplified indifference point, the largest value at which the participants chose the delayed option in the baseline items. For example, if an individual chose US$500 immediately over US$550 in 12 months, but US$600 in 12 months over US$500 immediately, then US$600 was the indifference value for subsequent scenarios. Those choices were then between US$500 in 12 months versus US$600 in 24 months (present bias), US$500 immediately versus US$700 in 24 months (subadditivity) and either being willing to wait 12 months for an additional US$100 in one set or being willing to lose US$100 to receive a reward now rather than in 12 months (delay–speedup). For consistency, the values were initially derived from local average income (local currency) and then from constant proportions based on the initial values (Supplementary Information). This approach was chosen over directly converting fixed amounts in each country due to the substantial differences in currencies and income standards.

Participants answered four additional questions related to the choice anomalies (gain–loss and magnitude effects were already collected in the first three sets). Due to contingencies in the instrument, all participants were then shown a present bias scenario (choice between 12 months and 24 months) followed by a subadditivity scenario (choice between immediate and 24 months). They were then randomly presented one of two delay–speedup scenarios (one framed as a bonus to wait, the other stated as a reduction to receive the reward earlier). After two similar but general choice and risk measures, they were presented with the second delay–speedup scenario. Due to the similarity in their wording, these scenarios were anticipated to have the lowest rates of anomalous choice. Finally, participants answered ten questions on financial circumstances, (simplified) risk preference, outlook and demographics. Participants could choose between the local official language (or languages) and English. By completion, 61 countries (representing approximately 76% of the world population) had participated.

We assessed temporal choice patterns in three ways. First, we tested discounting patterns from three baseline scenarios to determine preference for immediate or delayed choices for gains (at two magnitudes) and losses (one). Second, we analysed the prevalence of all choice anomalies using three additional items. Finally, with this information, we computed a discounting score based

on responses to all choice items and anomalies, which ranged from 0 (always prefer delayed gains or earlier losses) to 19 (always prefer immediate gains or delayed losses).

**Deviations from the pre-registered method.** There were minor deviations from the pre-registered method in terms of procedure. First, we did include an attention check, and the statement that we would not should have been removed; this was an error. Second, we had initially not planned to include students in the main analyses. Still, our recruitment processes turned out to be generally appropriate in terms of engaging students (16%) and non-students (84%) in the sample. We are therefore not concerned about skew and instead consider this a critical population. The impact of these deviations in the analyses is explained in the Supplementary Information.

**Statistical analysis.** Hierarchical generalized additive models[36] were estimated using fast restricted maximum likelihood and penalized cubic splines[56]. We selected the shrinkage version of cubic splines to avoid overfitting and foster the selection of only the most relevant nonlinear smooths[57]. Robustness checks were performed for the selection of knots (Supplementary Fig. 10) and spline basis (Supplementary Table 7), leaving the results unchanged. In these models, we estimated all effects of continuous variables as smooths to identify potential nonlinear variables, plus country of residence as random effects.

Relevant nonlinear effects were incorporated into our main linear and generalized mixed models. These models were fitted using a restricted maximum likelihood. Model convergence and assumptions were visually inspected. Bayesian versions of these models were estimated using four chains with 500 warmups and 1,000 iteration samples (4,000 total samples). We confirmed that all parameters presented $\hat{R}$ values equal to or below 1.01 and tail effective sample sizes above 1,000. We set the average proposal acceptance probability (delta) to 0.90 and the maximum tree depth to 15 (ref. [58]) to avoid divergent transitions. We employed a set of weakly informative priors, including $t$ distributions with three degrees of freedom and a standard deviation of 10 for model intercept and random effect standard deviations, a normal distribution with a zero mean, and a standard deviation of three for the fixed effect regression coefficient. For the standard deviation of the smooth parameter, we employed an exponential distribution with a rate parameter of one[59].

For smooth terms, we analysed whether each term was significant for the generalized additive model and presented substantial variance in the final models. We explored 95% confidence/credibility intervals for fixed effects[58] and examined support for potential null effects. All reported tests were two-tailed. Our power estimation considered unstandardized fixed regression effects of |0.15| and |0.07| as ultra-low effect sizes (categorical and continuous variables). Thus, assuming a null effect of a similar or lower magnitude (|0.10|), we computed log Bayes factors to quantify evidence favouring null effects of this range[60]. To understand the sensitivity of our results, we explored support for narrower null effects (ranges of |0.05| and |0.01|). As Bayes factors depend on prior specification, we also estimated the percentage of posterior samples within these regions (which could be understood as a region of practically equivalence analysis[61]). Both statistics provide sensitive, complementary evidence of whether null effects were supported or not[60,61]. Unfortunately, such analyses could not be conducted for smooth effects, as no single parameter could resume the relationship between the predictor and the dependent variable.

The analyses were conducted in R v.4.0.2 (ref. [62]) using the Microsoft R Open distribution[63]. The meta-analyses were conducted using the meta package. Nonlinear effects were studied using the mgcv[64] package, with the main models being estimated using the gamm4 (ref. [65]) and the brms[58] packages for frequentist and Bayesian estimation, respectively. All graphs were created using the ggplot2 (ref. [66]) (v.3.3.3) package. Data manipulations were conducted using the tidyverse[67] family of packages (v.1.3.0).

**Deviation from the pre-registered plan.** We aimed to follow our pre-registration analyses as closely as possible. On certain occasions, we decided to amplify the scope of the analyses and present robustness checks for the results presented by employing alternative estimation and inference techniques.

There was only one substantive deviation from our pre-registered analyses aside from the delay–speedup calculation. In the original plan, we intended to explore the role of financial status. In our final analysis, we employed individual assets and debts to this end. Assets and debts were included as raw indicators instead of inequality measures because we did not find reliable national average assets or individual debt sources.

One minor adaptation from our pre-registration involved our plan to test for nonlinear effects and use Bayesian estimation only as part of our exploratory analyses. However, as we identified several relevant nonlinear effects, we modified our workflow to accommodate those as follows: (1) we initially explored nonlinear effects using hierarchical generalized additive (mixed) models, (2) we included relevant nonlinear effects in our main pre-registered models and (3) we estimated Bayesian versions of these same models to test whether null effects could be supported in certain cases.

**Reporting summary.** Further information on research design is available in the Nature Research Reporting Summary linked to this article.

## Data availability
All data will be posted at https://osf.io/njd62 on September 1, 2022, while additional work is completed on an interactive tool with these data. Prior to this date, the data are available on request. Source data are provided with this paper.

## Code availability
All code will be posted at https://osf.io/njd62 on September 1, 2022, while additional work is completed on an interactive tool with these data. Prior to this date, the code is available on request.

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

## Acknowledgements

The authors received no specific funding for this work. A small amount of discretionary funding provided by K.R.'s institution paid for the pilot study participants and for honoraria to organizations that assisted with data collection in several locations. These were provided by Columbia University Undergraduate Global Engagement and the Department of Health Policy and Management. Funds to support open-access publication were provided by the MRC-CBU at the University of Cambridge through a UKRI grant (UKRI-MRC grant no. MC_UU_00005/6). None of these funders had any role in or influence over design, data collection, analysis or interpretation. All collaborators contributed in a voluntary capacity. We thank the Columbia University Office for Undergraduate Global Engagement. We also thank X. Li and L. Njozela, as well as the Centre for Business Research in the Judge Business School at the University of Cambridge.

## Author contributions

Conceptualization: K.R. Methodology: K.R., A.P., E.G.-G. and M.Vdo. Project coordination and administration: K.R. and Ta.Du. Supervision: K.R., J.K.B.L., Ma.Fr., P.K., Jo.Raz., C.E.-S., L.W. and Z.Z. Writing: K.R., E.G.-G., A.P., R.S.R. and Ir.Sob. Advisory: A.P. and R.S.R. Instrument adaptation, translation, circulation and recruitment: all authors. Analysis and visualization: E.G.-G. and K.R.

## Competing interests

The authors declare no competing interests.

## Additional information

**Correspondence and requests for materials** should be addressed to Kai Ruggeri.

**Kai Ruggeri**[1,2] ✉, **Amma Panin**[3], **Milica Vdovic**[4], **Bojana Većkalov**[5], **Nazeer Abdul-Salaam**[1], **Jascha Achterberg**[6,7], **Carla Akil**[8], **Jolly Amatya**[9], **Kanchan Amatya**[10], **Thomas Lind Andersen**[11], **Sibele D. Aquino**[12,13], **Arjoon Arunasalam**[14], **Sarah Ashcroft-Jones**[15], **Adrian Dahl Askelund**[16,17], **Nélida Ayacaxli**[1], **Aseman Bagheri Sheshdeh**[18], **Alexander Bailey**[14], **Paula Barea Arroyo**[19], **Genaro Basulto Mejía**[20], **Martina Benvenuti**[21], **Mari Louise Berge**[22], **Aliya Bermaganbet**[23], **Katherine Bibilouri**[1,24], **Ludvig Daae Bjørndal**[17], **Sabrina Black**[25], **Johanna K. Blomster Lyshol**[26], **Tymofii Brik**[27], **Eike Kofi Buabang**[28], **Matthias Burghart**[29], **Aslı Bursalıoğlu**[30], **Naos Mesfin Buzayu**[31], **Martin Čadek**[32], **Nathalia Melo de Carvalho**[12,33], **Ana-Maria Cazan**[34], **Melis Çetinçelik**[35], **Valentino E. Chai**[36], **Patricia Chen**[36], **Shiyi Chen**[37], **Georgia Clay**[38], **Simone D'Ambrogio**[15], **Kaja Damnjanović**[39], **Grace Duffy**[14], **Tatianna Dugue**[1], **Twinkle Dwarkanath**[1], **Esther Awazzi Envuladu**[40], **Nikola Erceg**[41], **Celia Esteban-Serna**[19], **Eman Farahat**[42,43], **R. A. Farrokhnia**[1], **Mareyba Fawad**[1], **Muhammad Fedryansyah**[44], **David Feng**[1,45], **Silvia Filippi**[46], **Matías A. Fonollá**[18], **René Freichel**[5], **Lucia Freira**[47], **Maja Friedemann**[15], **Ziwei Gao**[19], **Suwen Ge**[1], **Sandra J. Geiger**[48], **Leya George**[19], **Iulia Grabovski**[34], **Aleksandra Gracheva**[1,24], **Anastasia Gracheva**[1,49], **Ali Hajian**[50], **Nida Hasan**[1,24], **Marlene Hecht**[51,52], **Xinyi Hong**[53], **Barbora Hubená**[54], **Alexander Gustav Fredriksen Ikonomeas**[17], **Sandra Ilić**[39], **David Izydorczyk**[55], **Lea Jakob**[56,57], **Margo Janssens**[58], **Hannes Jarke**[6], **Ondřej Kácha**[6,59], **Kalina Nikolova Kalinova**[60], **Forget Mingiri Kapingura**[61], **Ralitsa Karakasheva**[62], **David Oliver Kasdan**[63], **Emmanuel Kemel**[64], **Peggah Khorrami**[65], **Jakub M. Krawiec**[66], **Nato Lagidze**[1], **Aleksandra Lazarević**[39], **Aleksandra Lazić**[39], **Hyung Seo Lee**[67], **Žan Lep**[68], **Samuel Lins**[69], **Ingvild Sandø Lofthus**[17], **Lucía Macchia**[70], **Salomé Mamede**[69],

Metasebiya Ayele Mamo[31], Laura Maratkyzy[71], Silvana Mareva[6], Shivika Marwaha[72], Lucy McGill[73], Sharon McParland[14], Anișoara Melnic[34], Sebastian A. Meyer[74,75], Szymon Mizak[66], Amina Mohammed[76], Aizhan Mukhyshbayeva[77], Joaquin Navajas[47,78], Dragana Neshevska[79], Shehrbano Jamali Niazi[80], Ana Elsa Nieto Nieves[81], Franziska Nippold[5], Julia Oberschulte[82], Thiago Otto[1], Riinu Pae[19], Tsvetelina Panchelieva[83], Sun Young Park[1], Daria Stefania Pascu[46], Irena Pavlović[39], Marija B. Petrović[39], Dora Popović[84], Gerhard M. Prinz[85], Nikolay R. Rachev[86], Pika Ranc[68], Josip Razum[84], Christina Eun Rho[1], Leonore Riitsalu[87], Federica Rocca[14], R. Shayna Rosenbaum[88,89], James Rujimora[90], Binahayati Rusyidi[44], Charlotte Rutherford[6], Rand Said[14], Inés Sanguino[15], Ahmet Kerem Sarikaya[1], Nicolas Say[91], Jakob Schuck[48], Mary Shiels[14], Yarden Shir[92], Elisabeth D. C. Sievert[93], Irina Soboleva[31], Tina Solomonia[94], Siddhant Soni[95], Irem Soysal[1,15], Federica Stablum[6,96], Felicia T. A. Sundström[97], Xintong Tang[1], Felice Tavera[98], Jacqueline Taylor[1], Anna-Lena Tebbe[99], Katrine Krabbe Thommesen[100], Juliette Tobias-Webb[101], Anna Louise Todsen[25], Filippo Toscano[46], Tran Tran[95], Jason Trinh[1], Alice Turati[1,24], Kohei Ueda[102], Martina Vacondio[103], Volodymyr Vakhitov[27], Adrianna J. Valencia[1,90], Chiara Van Reyn[28], Tina A. G. Venema[104], Sanne E. Verra[105], Jáchym Vintr[56,59], Marek A. Vranka[56], Lisa Wagner[106], Xue Wu[102], Ke Ying Xing[107], Kailin Xu[14], Sonya Xu[1,6], Yuki Yamada[102], Aleksandra Yosifova[108], Zorana Zupan[39] and Eduardo García-Garzon[109]

[1]Columbia University, New York, NY, USA. [2]Centre for Business Research, Judge Business School, University of Cambridge, Cambridge, UK. [3]UC Louvain, Louvain, Belgium. [4]Faculty of Media and Communications, Belgrade, Serbia. [5]University of Amsterdam, Amsterdam, the Netherlands. [6]University of Cambridge, Cambridge, UK. [7]MRC Cognition and Brain Sciences Unit, Cambridge, UK. [8]American University of Beirut, Beirut, Lebanon. [9]UN Major Group for Children and Youth (UNMGCY), Kathmandu, Nepal. [10]United Nations Children's Fund (UNICEF), Kathmandu, Nepal. [11]PPR Svendborg, Svendborg, Denmark. [12]Pontifical Catholic University of Rio de Janeiro, Rio de Janeiro, Brazil. [13]Laboratory of Research in Social Psychology, Rio de Janeiro, Brazil. [14]Queen's University Belfast, Belfast, UK. [15]University of Oxford, Oxford, UK. [16]Nic Waals Institute, Oslo, Norway. [17]University of Oslo, Oslo, Norway. [18]St. Lawrence University, Canton, NY, USA. [19]University College London, London, UK. [20]Centro de Investigación y Docencias Económicas, Ciudad de México, México. [21]University of Bologna, Bologna, Italy. [22]Unaffiliated, Budapest, Hungary. [23]Workforce Development Center, Nur-Sultan, Kazakhstan. [24]Sciences Po, Paris, France. [25]University of St Andrews, St Andrews, UK. [26]Oslo New University College, Oslo, Norway. [27]Kyiv School of Economics, Kyiv, Ukraine. [28]KU Leuven, Leuven, Belgium. [29]University of Konstanz, Konstanz, Germany. [30]Loyola University Chicago, Chicago, IL, USA. [31]Duke Kunshan University, Kunshan, China. [32]Leeds Beckett University, Leeds, UK. [33]Estácio de Sá University, Rio de Janeiro, Brazil. [34]Transilvania University of Brasov, Brasov, Romania. [35]Max Planck Institute for Psycholinguistics, Nijmegen, the Netherlands. [36]National University of Singapore, Singapore, Singapore. [37]The University of Hong Kong, Hong Kong SAR, China. [38]Technische Universität Dresden, Dresden, Germany. [39]University of Belgrade, Belgrade, Serbia. [40]University of Jos, Jos, Nigeria. [41]University of Zagreb, Zagreb, Croatia. [42]Ain Shams University, Cairo, Egypt. [43]International Socioeconomics Laboratory, New York, NY, USA. [44]Universitas Padjadjaran, Bandung, Indonesia. [45]London School of Economics and Political Science, London, UK. [46]University of Padua, Padua, Italy. [47]Universidad Torcuato Di Tella, Buenos Aires, Argentina. [48]University of Vienna, Vienna, Austria. [49]The Wharton School of the University of Pennsylvania, Philadelphia, PA, USA. [50]University of Tehran, Tehran, Iran. [51]Max Planck Institute for Human Development, Berlin, Germany. [52]Humboldt University of Berlin, Berlin, Germany. [53]Duke University, Durham, NC, USA. [54]Unaffiliated, Prague, Czech Republic. [55]University of Mannheim, Mannheim, Germany. [56]Charles University, Prague, Czech Republic. [57]National Institute of Mental Health, Klecany, Czech Republic. [58]Tilburg University, Tilburg, the Netherlands. [59]Green Dock, Hostivice, Czech Republic. [60]Leiden University, Leiden, the Netherlands. [61]University of Fort Hare, Alice, South Africa. [62]Unaffiliated, London, UK. [63]Sungkyunkwan University, Seoul, Republic of Korea. [64]GREGHEC, CNRS, HEC Paris, Jouy en Josas, France. [65]Harvard University, Boston, MA, USA. [66]SWPS University of Social Sciences and Humanities, Warsaw, Poland. [67]Emory University, Atlanta, GA, USA. [68]University of Ljubljana, Ljubljana, Slovenia. [69]University of Porto, Porto, Portugal. [70]Harvard Kennedy School, Cambridge, MA, USA. [71]Nazarbayev University, Nur-Sultan, Kazakhstan. [72]University College Cork, Cork, Ireland. [73]University of Groningen, Groningen, the Netherlands. [74]Fundación Paraguaya, Asunción, Paraguay. [75]Colmena, Asunción, Paraguay. [76]Gombe State University, Gombe, Nigeria. [77]University of Chicago, Chicago, IL, USA. [78]National Scientific and Technical Research Council, Buenos Aires, Argentina. [79]Ss. Cyril and Methodius University, Skopje, North Macedonia. [80]McGill University, Montreal, Quebec, Canada. [81]Universidad Autónoma de Madrid, Madrid, Spain. [82]Ludwig-Maximilians-Universität München, Munich, Germany. [83]IPHS—Bulgarian Academy of Sciences, Sofia, Bulgaria. [84]Ivo Pilar Institute of Social Sciences, Zagreb, Croatia. [85]Bezirkskrankenhaus Straubing, Straubing, Germany. [86]Sofia University St. Kliment Ohridski, Sofia, Bulgaria. [87]University of Tartu, Tartu, Estonia. [88]York University, Toronto, Ontario, Canada. [89]Rotman Research Institute, Baycrest, Toronto, Ontario, Canada. [90]University of Central Florida, Orlando, FL, USA. [91]Prague University of Economics and Business, Prague, Czech Republic. [92]Tel Aviv University, Tel Aviv, Israel. [93]Helmut Schmidt University, Hamburg, Germany. [94]Tbilisi State University, Tbilisi, Georgia. [95]Erasmus University Rotterdam, Rotterdam, Netherlands. [96]University of Trento, Trento, Italy. [97]Uppsala University, Uppsala, Sweden. [98]University of Cologne, Cologne, Germany. [99]Max Planck Institute for Human Cognitive and Brain Sciences, Leipzig, Germany. [100]Copenhagen University, Copenhagen, Denmark. [101]Kaplan Business School, Sydney, New South Wales, Australia. [102]Kyushu University, Fukuoka, Japan. [103]University of Klagenfurt, Klagenfurt, Austria. [104]Aarhus University, Aarhus, Denmark. [105]Utrecht University, Utrecht, the Netherlands. [106]University of Zurich, Zurich, Switzerland. [107]Cornell University, Ithaca, NY, USA. [108]New Bulgarian University, Sofia, Bulgaria. [109]Universidad Camilo José Cela, Madrid, Spain. ✉e-mail: kai.ruggeri@columbia.edu

# Reporting Summary

## Statistics

For all statistical analyses, confirm that the following items are present in the figure legend, table legend, main text, or Methods section.

| n/a | Confirmed | |
|---|---|---|
| ☐ | ☒ | The exact sample size (*n*) for each experimental group/condition, given as a discrete number and unit of measurement |
| ☐ | ☒ | A statement on whether measurements were taken from distinct samples or whether the same sample was measured repeatedly |
| ☐ | ☒ | The statistical test(s) used AND whether they are one- or two-sided<br>*Only common tests should be described solely by name; describe more complex techniques in the Methods section.* |
| ☐ | ☒ | A description of all covariates tested |
| ☐ | ☒ | A description of any assumptions or corrections, such as tests of normality and adjustment for multiple comparisons |
| ☐ | ☒ | A full description of the statistical parameters including central tendency (e.g. means) or other basic estimates (e.g. regression coefficient) AND variation (e.g. standard deviation) or associated estimates of uncertainty (e.g. confidence intervals) |
| ☐ | ☒ | For null hypothesis testing, the test statistic (e.g. *F*, *t*, *r*) with confidence intervals, effect sizes, degrees of freedom and *P* value noted<br>*Give P values as exact values whenever suitable.* |
| ☐ | ☒ | For Bayesian analysis, information on the choice of priors and Markov chain Monte Carlo settings |
| ☐ | ☒ | For hierarchical and complex designs, identification of the appropriate level for tests and full reporting of outcomes |
| ☐ | ☒ | Estimates of effect sizes (e.g. Cohen's *d*, Pearson's *r*), indicating how they were calculated |

*Our web collection on statistics for biologists contains articles on many of the points above.*

## Software and code

Policy information about availability of computer code

| | |
|---|---|
| Data collection | Data collection was conducted using the Qualtrics XM web service platform. |
| Data analysis | All code relative to power estimation and pre-registration is openly available in https://osf.io/jfvh4 with all data stored at https://osf.io/njd62/. All analyses were conducted in R 4.0.2 using the Microsoft R Open Distribution 4.0.2. Principal packages employed were tidyverse (1.3.0) for data handling, meta (4.18-2) for estimating meta-analyses, mgcv (1.8-31) for estimating hierarchical generalized additive models, gamm4 (0.2-6) for estimating mixed linear and generalized models, and brms (2.14.4) for computing the Bayesian version of the latter. All visualizations were created using ggplot2 (3.3.3). Analysis code and data will be publicly prior to publication in the same link. |

For manuscripts utilizing custom algorithms or software that are central to the research but not yet described in published literature, software must be made available to editors and reviewers. We strongly encourage code deposition in a community repository (e.g. GitHub). See the Nature Portfolio guidelines for submitting code & software for further information.

## Data

Policy information about availability of data

All manuscripts must include a data availability statement. This statement should provide the following information, where applicable:
- Accession codes, unique identifiers, or web links for publicly available datasets
- A description of any restrictions on data availability
- For clinical datasets or third party data, please ensure that the statement adheres to our policy

All data will be made available at https://osf.io/njd62/ soon after publication. We originally intended to make it available on 1 September, 2022, but we will treat this as a latest-posting date, and make clear in the data availability that we will share it with researchers that request it beforehand. The additional data used from secondary sources are: 1. Country classifications: https://datahelpdesk.worldbank.org/knowledgebase/articles/906519-world-bank-country-and-lending-groups.

2. Gross domestic product (in current US$). Data obtained from World Bank database (https://data.worldbank.org/indicator/NY.GDP.MKTP.CD)
3. GINI index. World Bank estimate. We used the latest data available retrieved from https://data.worldbank.org/indicator/SI.POV.GINI
4. Inflation: We used inflation as relative in consumer prices index (change in annual percentage) from the World Bank database (retrieved from https://data.worldbank.org/indicator/FP.CPI.TOTL.ZG) 5. The stimulus data used in Figure 1 is not publicly released as it belongs to a financial institution. Inquiries about accessing this data may be sent to arf25@columbia.edu.

# Field-specific reporting

Please select the one below that is the best fit for your research. If you are not sure, read the appropriate sections before making your selection.

☐ Life sciences    ☒ Behavioural & social sciences    ☐ Ecological, evolutionary & environmental sciences

For a reference copy of the document with all sections, see nature.com/documents/nr-reporting-summary-flat.pdf

# Behavioural & social sciences study design

All studies must disclose on these points even when the disclosure is negative.

| | |
|---|---|
| Study description | A 61-country decision-study testing temporal discounting with an emphasis on economic inequality. All participants completed a survey of approximately 25-30 items, which was identical for all participants with a small number of contingent items. |
| Research sample | Entirely random sample of adults (locally-defined; typically 18 and older) from 61 countries (47% female; mean age = 34). Samples were not weighted or recruited in a way that ensured representativeness, but instead used the most random approach possible given the pandemic (i.e., all testing done online, typically on personal computers or at community centers in regions with low computer access). As explained in the next box, we targeted a sample of adults that would produce a sufficiently powered estimate for comparisons within and between countries. We only focused on adults due to the nature of the financial topics. |
| Sampling strategy | We use what we refer to as the Demic-Veckalov (named for Emir Demic and Bojana Veckalov) method for sampling: All collaborators used a range of circulation points, including email lists, discussion boards, and social media pages to recruit as random a sample as possible. This meant we primarily did not use individual pages to recruit, but instead, found recent posts with high engagement (often related to financial news) as well as common-interest platforms (e.g., Reddit channels). We also contacted NGOs and other organizations to assist with circulation. As described in the preregistration (https://osf.io/jfvh4), we identified a minimum sample size of 30 to achieve sufficient power (.95) for extremely small effects, though we aimed for over 120 participants as a minimum target for each country. The minimum of 30 was easily achieved for all countries included in the final version; a small number of countries did not meet the ideal 120. We also used a sliding scale target of 240 for countries of over 10 million population, and 360 for those over 100 million. |
| Data collection | All participants completed the study via Qualtrics; no researcher was present at the time of data collection and there were no conditions for blinding. Participants could choose the local national language or English (in some cases, additional languages were offered). No hard-copy versions were used. In Nepal and Ethiopia, we were informed that a community center may have hosted participants to support data collection, but this was organized outside the research team. |
| Timing | All surveys were collected between late July and early September 2021. |
| Data exclusions | We removed 6,141 participants (23.7%) who did not pass the attention check. 69 participants were removed for giving nonsensical responses to open data text (i.e., "helicopter" as gender). We removed 13 participants claiming to be over 100 years old. Based on the length of time for responses, 5,870 individuals that completed faster than three times the absolute deviations below the median time or that took less than 120 seconds to respond were removed. We further removed responses from IP addresses identified as either "tests" or "spam" by the Qualtrics service (264). |
| Non-participation | 9,434 individuals did not complete at least 90% of the survey and were therefore excluded. |
| Randomization | No randomization was used apart from a small number of specific questions in the survey. All participants completed essentially the same version of the instrument. |

# Reporting for specific materials, systems and methods

We require information from authors about some types of materials, experimental systems and methods used in many studies. Here, indicate whether each material, system or method listed is relevant to your study. If you are not sure if a list item applies to your research, read the appropriate section before selecting a response.

## Materials & experimental systems

| n/a | Involved in the study |
|---|---|
| ☒ | ☐ Antibodies |
| ☒ | ☐ Eukaryotic cell lines |
| ☒ | ☐ Palaeontology and archaeology |
| ☒ | ☐ Animals and other organisms |
| ☐ | ☒ Human research participants |
| ☒ | ☐ Clinical data |
| ☒ | ☐ Dual use research of concern |

## Methods

| n/a | Involved in the study |
|---|---|
| ☒ | ☐ ChIP-seq |
| ☒ | ☐ Flow cytometry |
| ☒ | ☐ MRI-based neuroimaging |

# Human research participants

Policy information about studies involving human research participants

| | |
|---|---|
| Population characteristics | Participants were 47% female with a mean age of 34. Almost 100% of participants had completed some formal education, with 72% completing some form of higher education. 16% were current students. Over half (53%) had full-time employment; 10% were unemployed and 3% were retired. (See above for more.) |
| Recruitment | We use what we refer to as the Demic-Veckalov (named for Emir Demic and Bojana Veckalov) method for sampling: All collaborators used a range of circulation points, including email lists, discussion boards, and social media pages to recruit as random a sample as possible. This meant we primarily did not use individual pages to recruit, but instead, found recent posts with high engagement (often related to financial news) as well as common-interest platforms (e.g., Reddit channels). We also contacted NGOs and other organizations to assist with circulation. The primary forms of bias that this could create would be over-representation of individuals with computers/social media accounts, younger and more educated participants (due to the types of news stories often used as a conduit for recruiting), and individuals that speak the primary local language. |
| Ethics oversight | The study was approved by the Institutional Review Board at Columbia University in the City of New York. |

Note that full information on the approval of the study protocol must also be provided in the manuscript.

