## [Peer Review File · Nature Human Behaviour]

Peer Review Information

Journal: Nature Human Behaviour

Manuscript Title: The globalizability of temporal discounting

Corresponding author name(s): Kai Ruggeri

Reviewer Comments & Decisions:

Decision Letter, initial version:

8th February 2022

Dear Kai,

Thank you once again for your manuscript, entitled "The globalizability of temporal discounting," and for your patience during the peer review process.

Your manuscript has now been evaluated by 4 reviewers, whose comments are included at the end of this letter. Although the reviewers find your work to be of interest, they also raise some important concerns. We are interested in the possibility of publishing your study in Nature Human Behaviour, but would like to consider your response to these concerns in the form of a revised manuscript before we make a decision on publication.

To guide the scope of the revisions, the editors discuss the referee reports in detail within the team, including with the chief editor, with a view to (1) identifying key priorities that should be addressed in revision and (2) overruling referee requests that are deemed beyond the scope of the current study. We hope that you will find the prioritised set of referee points to be useful when revising your study. Please do not hesitate to get in touch if you would like to discuss these issues further.

In particular, your revision must address the following (as well as all other reviewer comments):

1. Address concerns raised by Reviewers 1 and 3 regarding the attention check and exclusion criteria, and carry out the sensitivity analysis recommended by Reviewer 1.

2. Address reviewer concerns about the strength of the introduction and clarity of your study's purpose, as well as the strength of the interpretations of results.

3. Provide evidence of the reliability and validity of your temporal discounting score, as requested by Reviewer 1.

4. Ensure that results from correlational analyses are not discussed causally or applied to answer causal/directional research questions.

5. Fully and transparently discuss the limitations of your work, including the lack of task incentivization.

6. Ensure that all analysis code is accurate and runs correctly, and that data are fully anonymized.

In sum, we invite you to revise your manuscript taking into account all reviewer and editor comments. We are committed to providing a fair and constructive peer-review process. Do not hesitate to contact us if there are specific requests from the reviewers that you believe are technically impossible or unlikely to yield a meaningful outcome.

We hope to receive your revised manuscript within three months. We understand that the COVID-19 pandemic is causing significant disruption for many of our authors and reviewers. If you cannot send your revised manuscript within this time, please let us know - we will be happy to extend the submission date to enable you to complete your work on the revision.

- Include a "Response to the editors and reviewers" document detailing, point-by-point, how you addressed each editor and referee comment. If no action was taken to address a point, you must provide a compelling argument. This response will be used by the editors to evaluate your revision and sent back to the reviewers along with the revised manuscript.
- Highlight all changes made to your manuscript or provide us with a version that tracks changes.

[REDACTED]

We look forward to seeing the revised manuscript and thank you for the opportunity to review your work. Please do not hesitate to contact me if you have any questions or would like to discuss these revisions further.

Sincerely,
Aisha

Aisha Bradshaw, PhD
Senior Editor
Nature Human Behaviour

Reviewer expertise:

Reviewer #1: social psychology, temporal discounting

Reviewer #2: behavioural economics, statistics

Reviewer #3: decision science, Bayesian modelling

Reviewer #4: behavioural economics

REVIEWER COMMENTS:

Reviewer #1:
Remarks to the Author:

First of all, I'd like to thank the authors for this interesting submission. The main topic of this paper - exploring how temporal discounting and its anomalies are related to economic factors and what extent these relationships are generalizable across culturally diverse populations (61 countries) - fits the scope of Nature Human Behaviour and is highly relevant for its broad readership. There are many positives to the paper. In general, the paper is well-written and its main messages, temporal discounting and its anomalies, are robust on a global level; living in a financially challenging environment is associated with higher temporal discounting - clearly expressed in the main text as well as in the abstract. The theoretical background is succinct and cites most of the literature relevant to the topic. The data collection procedure (including the a priori simulations available at the OSF), as well as its processing, were thoroughly designed. This is nicely documented in the supplementary materials and in the pre-registration. The analytic workflow is to a very high level and, as far as I was able to understand the calculations (I'm not an expert in GAM), I didn't notice any major mistakes while reviewing the functions/code. However, I wasn't able to properly run the code and reproduce the results but I'll get to that point later. The study was pre-registered and the authors transparently

disclose the deviations from the pre-registration. Despite the complexity of the analyses, the results are comprehensively written and visualized, and the interpretations (usually) logically follow the findings. All in all, I'm very pleased with the paper and I think that the authors did a great job. The findings and their robustness (not only the methodological/statistical rigor but also the cultural diversity of the sample) significantly advance the field of temporal discounting and its association with economic circumstances. Below, I provide some (mostly minor) comments and suggestions on how to improve the paper and supplementary materials.

Theoretical background and research questions

The theoretical background flows well and is succinct. The strengths and novelty of the authors' approach to addressing these problems are nicely summarized. However, I believe it would benefit from some restructuring in the main text and additional information in the supplementary materials. (1) In particular, I'd welcome a clearer paragraph on the aims of the research. While they are in the text, the information is scattered across three paragraphs and also blends in with the paragraph about inflation. A summary of these aims would help the reader grasp the idea of the paper right from the beginning. Maybe moving the information about assessing temporal discounting from the first paragraph to the beginning of the third paragraph on page 3 would be enough although perhaps formulating a paragraph that mentions time discounting, its anomalies and potential economic determinants would be a better choice. (2) It would also help to briefly mention the anomalies earlier in the text – for example, in the first paragraph on page 2. (3) The authors write that "This tendency, known as temporal discounting, is routinely associated with lower wealth" and support the statement with several references. Could the authors provide a brief, quantitative, synthesis of these (and maybe also other findings) on this topic as part of the supplementary materials? This is not necessary for the paper per se but it would be very informative to summarize the available evidence (maybe this would also help in setting informative priors in the Bayesian analyses if the authors or an interested reader decided to do further sensitivity analyses, although I guess it wouldn't change the results in this case).

Methods

As I wrote earlier, the data collection was well planned, and the research methodology was thoroughly designed. In order to assess statistical power, the authors ran a priori simulations for both GLM and HLM models. Most of the details are sufficiently explained in the supplementary materials. The authors follow the pre-registration (even though the pre-registered hypotheses were somewhat vague - resembling research questions rather than testable hypotheses) and transparently report the deviations. I still, however, have some questions/suggestions. (1) Could the authors clarify how the total temporal discounting score was computed? I understand that the authors combined all items ("...we computed a discounting score based on responses to all choice items and anomalies, which ranged from 0 to 19..."). However, I'm not convinced that such an approach delivers a reliable/valid score. Could the authors provide a rationale (and, ideally, include it in the supplementary materials) regarding the reliability, but mainly validity of this score? (2) Furthermore, the authors did a nice job regarding the data cleaning and set rational inclusion criteria. Some of their choices (e.g., removing participants who had completed the survey in less than 120 seconds; removing "extreme outliers larger than 100 times the median absolute deviation above the country median for income and 1000

times larger than the median absolute deviation for national median assets”) could be considered too liberal. I wonder to what extent the results would change if more conservative inclusion criteria were applied? Could the authors run a sensitivity analysis on this? (3) Besides the delay-speedup scenario, was any randomization concerning the measures/items applied? (4) The authors tested temporal discounting and its anomalies across different scenarios, which is (as noticed also by the authors themselves) a very strong point of this paper. As is typical for research on temporal discounting, researchers’ degrees of freedom play a huge role in creating the items (e.g., the usage of 10% of monthly household income; delayed values of, say, 150%; waiting period of one year, etc.). Could the authors discuss (in the supplementary materials) why they made these choices and how researchers’ degrees of freedom could potentially affect the results in this case? I know this might be a difficult request, so please just consider it as a suggestion to strengthen the robustness of the study. (5) The authors have used an interesting method of participant recruitment. Since this is a global study with a potentially significant impact on the field and the related generalizability of the results, could the authors provide a supplementary table which would compare how the recruited sample differs from the population of each country in basic demographic characteristics (i.e., gender, age, completed education; of course, if such data are available for the countries)? (6) One more thing - to preserve the participants’ anonymity, please remove variables like IP address or location from the raw dataset.

Results

The authors nicely extract and report the most relevant findings and enrich them with additional results in the supplementary materials. However, as there are so many analyses, it was occasionally hard to follow them in all documents (main paper, supplementary material, code) so I might be wrong in some of my following comments. However, I’ve noticed several things I’d like to share with the authors. (1) Marginal differences in p-values or ORs lead to substantial differences in the interpretations. For instance, the authors write that they found “...a positive effect (of GDP) for delay-speedup (OR = 0.95, 95% CI [0.91, 0.99], $p = 0.002$)” and in the next sentence they state “...we found no effect on the remaining anomalies ($0.95 < OR < 1.01$; $0.027 < p < 0.688$)”. I understand that this is based on p-values/CIs containing zero (and probably also on BFs), but the ORs are not really different. (2) On the other hand, on one occasion, the authors state they observed “...only modest variability (of temporal discounting) for national means”. I’m not sure if this is true - both relative (I^2) and absolute (τ^2) values of heterogeneity are (relatively) high. This can be nicely seen in the example provided by the authors who compared Japan ($M = 8.1$, $SD = 3.9$) and Argentina ($M = 15.1$, $SD = 3$), with the difference for these two countries being Cohen’s $d \approx 2$. Speaking of high heterogeneity, maybe the authors could try running a meta-regression to examine its source. (4) Following this, the authors conclude that “this is largely based on finding remarkable consistency and robustness in patterns of intertemporal choice across 61 countries, with substantially more variability within each country than between their means.”. I’d suggest a slight reformulation of this statement although maybe I just missed something, and the conclusion is correct.

Statistical analysis, code, and data

The analytic workflow described in the paper/supplementary material is precise, and the proposed statistical analyses are appropriately used and technically sound (at least to the extent I was able to

understand them), thus allowing for robust findings. However, I struggled while reviewing the code / trying to reproduce the findings. The code is not fully reproducible (I've tried it on two different machines as I was expecting some potential problems with RStan) and would need some adjustments and more detailed annotation. (1) First of all, I'd like the authors to check the code (push-button style) on different machines. (2) Please include the output of `sessionInfo()` in the script. (3) I was able to run file 2_1 with the sourced data (although with several errors) but not with data produced by file 1_1. The same applies to the graphs (3_1) and supplementary analyses (4_1). Besides the unreproducible parts, I've noticed two substantial typos: the function `fit_models_lmer` contains a mistake in line 571: `mg5 = mg6` should be `mg6 = mg6`; file 2_1, line 101 should be `cmdstanr` instead of `cmdrstan`. (4) More detailed annotation would be very much welcomed to understand better what's happening in the code. I'm aware that preparing a more annotated code is a difficult and unpleasant job. However, it allows for easier comprehension and easier reproducibility by independent researchers, which I find extremely important. (4) Please consider using standard scripts instead of RMarkdown files, although this is just a matter of preference.

Discussion/conclusion

The discussion is concise and summarizes the main findings and contributions of the study. Although I enjoy reading short yet succinct introductions and discussions, I believe that some adjustments could strengthen the study even more. (1) I'm missing a paragraph about the limitations of the study (e.g., naturally, there's a difference between hypothetical and real scenarios, although evidence on the correlation of the two differs) and perhaps also a few additional sentences on the robustness of the study. (2) The authors also depict some possible implications (e.g., for policymaking) but don't elaborate on that any further. Covering these issues in more depth (not necessarily in the main text – these suggestions could be part of the supplementary materials) could make the paper even more robust and comprehensive.

Very minor comments

(1) In the introduction, the authors write "To test our hypotheses...". Besides one exception in the paragraph about the effects of inflation on temporal discounting, the paper doesn't contain any hypotheses (instead, it contains goals of the research), so reformulation would help. (2) In the first sentence of the last paragraph on page 4, please add that it was based on average income per month. (3) The final dataset (the dataset on which the analyses were performed; `dat_filter`) consists of a sample of 13,324 (becomes 13,373 when I run file 1_1 instead of loading `1_2_data_files.RData` provided by the authors) instead of the 13,629 participants the authors mention in the text.

My final remarks

This study has the potential to be one of the most influential papers on the topic of temporal discounting. Once again, I'd like to thank the authors for this submission, and I hope they'll find some of my comments and suggestions useful.

All the best,

Matúš Adamkovič

Reviewer #2:

Remarks to the Author:

I am positive towards this work. I believe that the dataset presented in the paper is extraordinary and that a great effort was devoted to produce this piece of research. I also believe that the main conclusion that temporal discounting is a global phenomenon, is very relevant to the literature. I have just a few comments on this work:

1) The abstract is very focused on the finding that economic inequality is a predictor of time preferences. However, I am not sure this is the main message which you get from the paper. Plus, economic inequality is correlated to so many other outcomes. I would tone down all those places where you attribute your finding to economic inequality and clearly state it is very correlational.

2) The structure of the introduction is not so clear. Some findings are highly emphasized (like inflation) at a point where I would have expected the focus to be on other points. Plus, I am not sure that the spending pattern of the CARES Act is a necessary justification for your work. Most importantly, you give the impression that you are going to focus a lot on individual financial conditions, which is not so true in the rest of the paper.

3) In the conclusion, you highlight that you find consistency and robustness of the intertemporal anomalies. I think this is the most robust point and the main innovation of your work.

4) To sum up, it is not clear what your main focus is. I would say the point in 3, since the conclusion section seems quite clear and structured. I invite you to harmonize abstract/introduction and conclusion. Plus, I think it would be beneficial having a clear roadmap of the hypotheses that you want to test and a table which summarizes which hypotheses are supported by the data. It is quite difficult getting the main messages out of your work.

5) I think that that this work is too premature to claim that "It also disrupts any notion that lower-income individuals are somehow concerning decision-makers, as negative environments are widely influential." There are not many analyses referring to individual financial conditions. Plus, when you refer to individual financial conditions, I would use a relative measure of wealth and not an absolute one like assets. I suggest you show how individuals ranking at a given percentile of the income distribution (for example) behave, depending on how unequal their country of origin is. Until that point, I think your conclusion is premature and you should tone it down.

6) Inflation has much more focus in the introduction than in the rest of the paper. Plus, do you have reasons to expect a difference due to only very high inflation?

7) There are several studies which measured time preferences at an almost global level. You only refer to Falk et al. (2018). It would be interesting to see a comparison of your data with all these other data. However, I realize that your data are more detailed and that they will probably be the gold standard for the future. Other papers include: Burro et al. (2022) Patience Decreases With Age for the Poor but Not for the Rich: An International Comparison (JEBO); Wang et al. (2016) How time preferences differ: Evidence from 53 countries (JOEP); Rieger et al. (2021) Universal time preference (Plos One). In particular, in the last paper (the Plos One), the authors list a series of global measures of time preferences and they aggregate them into a single one.

8) Some sentences seem incomplete (description of fig 4 for example) and I think that there are a few typos ("evaluated a variety of Bayesian approaches" for example).

Reviewer #3:
Remarks to the Author:

In this paper, the authors present a very large dataset examining intertemporal choice phenomena across 61 countries and over 90 participants in almost every country studied. By analyzing the choice proportions across several scenarios, designed to elicit overall patterns of discounting as well as several important anomalies of intertemporal choice, they show that these anomalies largely appear consistently across countries, but that these anomalies – and especially overall patterns of discounting – vary according to both individual and country-level income as well as indices related to inequality (GINI), inflation, and wealth / assets.

My immediate impression is that this is among the largest and arguably most important data sets ever collected on intertemporal choice. The paper is clearly written and provides a straightforward set of analyses that highlight the main findings. Most of the issues and recommendations I have are relatively minor and aimed at making sure that the manuscript is as clear as possible and reaches a wide audience.

The biggest question I had while reading was: What was the attention check that was added to the study? This was a deviation from the pre-registration, as the authors note, but I don't see anywhere that the nature of the attention check was discussed. I know that sometimes in intertemporal choice work a "dominated" option is offered (e.g., \$100 now or \$50 in a year), and it could be the case that the prevalence of this type of responding across countries is theoretically interesting. But I could not find the attention check items in the supplement or anywhere in the paper, so it is hard to say whether they would be interesting or not. Given that over 6000 participants failed the attention check, it seems like this could be an important issue (as well as its translation across languages).

Another issue that is likely to come up with some readers is the fact that participants were not compensated for their participation (except in the Japanese sample?), which may be taken by many (especially economists) to mean that participants were not motivated to exhibit rational / optimal

behavior. It may be worth heading off this criticism in the paper. Here are some papers that may help in doing so:

Wiseman, D. B., & Levin, I. P. (1996). Comparing risky decision making under conditions of real and hypothetical consequences. *Organizational behavior and human decision processes*, 66(3), 241-250.

Kühberger, A., Schulte-Mecklenbeck, M., & Perner, J. (2002). Framing decisions: Hypothetical and real. *Organizational Behavior and Human Decision Processes*, 89(2), 1162-1175.

Amlung, M., & MacKillop, J. (2015). Further evidence of close correspondence for alcohol demand decision making for hypothetical and incentivized rewards. *Behavioural processes*, 113, 187-191.

Madden, G. J., Begotka, A. M., Raiff, B. R., & Kastern, L. L. (2003). Delay discounting of real and hypothetical rewards. *Experimental and clinical psychopharmacology*, 11(2), 139.

Locey, M. L., Jones, B. A., & Rachlin, H. (2011). Real and hypothetical rewards. *Judgment and Decision making*, 6(6), 552.

It has also become more common in intertemporal choice literature to look at not just indifference points but generative model parameters related to discounting. There are a few reasons for this, including the observation that these parameters provide a more reliable and (at least predictively) valid account of behavior on intertemporal choice tasks. This can naturally be improved with methods like hierarchical Bayesian estimation of discounting rates (k) and/or more sophisticated models that can predict all of the anomalies that are mentioned in the paper (e.g., direct difference model; Dai Busemeyer, 2014).

Odum, A. L. (2011). Delay discounting: I'm ak, you're ak. *Journal of the Experimental Analysis of Behavior*, 96(3), 427-439.

Dai, J., & Busemeyer, J. R. (2014). A probabilistic, dynamic, and attribute-wise model of intertemporal choice. *Journal of Experimental Psychology: General*, 143(4), 1489.

Molloy, M. F., Romeu, R. J., Kvam, P. D., Finn, P. R., Busemeyer, J., & Turner, B. M. (2020). Hierarchies improve individual assessment of temporal discounting behavior. *Decision*, 7(3), 212.

I am not suggesting that the authors need to carry out sophisticated model-based analyses of the results (although this would certainly be interesting!), but it is worth mentioning alternative approaches to data analysis on intertemporal choice tasks that might elucidate other interesting individual or country-level differences in behavior. This could even be spun as a future research direction that will be enabled by making the data set openly available, as the authors apparently plan to do.

Reviewer #4:
Remarks to the Author:

This is a great paper. The design and analysis choices are excellent and follow the current standard. For example, I'm happy that they report the raw number of SS vs LL choices (reversed for losses), rather than calculating discount rates with a hyperbolic model or similar.

Here are my suggestions for improvement:

- I'm not sure what the policy is at this journal, but it would be great if you would publicly share the (deidentified) data and the data analysis scripts from this project, via something like <http://osf.io>. As you point out, there has been a replication crisis and crisis of confidence in behavioral sciences, and we need transition to open science standards.
- although I love the # of SS choices for the primary analyses and reporting, it would also be interesting and useful to see them converted to annualized discount rates (using the continuously compounded exponential model) for comparison with other studies and economic conditions. A table that shows the annualized discount rate as a percentage in each country and each condition would be nice, similar to the Loewenstein & Prelec (1992) paper.
- "anyone facing a negative financial environment – even with better incomes within that environment – is likely to make decisions that prioritize immediate clarity over future uncertainty" -- are you studying time preferences, or risk preferences? I would appreciate a deeper exploration of time, risk, and why you observe these results.
- A methodological limitation of the paper is that all choices are hypothetical. This makes it easier to run these study across so many different locations and teams (it's logistically very challenging to do intertemporal choices "for real"), but it does mean we have less confidence in the results, as we aren't sure if differences observed between countries and individuals are due to differences in reporting or "real" differences in time preferences. I think you should be more upfront (in the abstract, etc) about the fact that these are hypothetical choices.
- The axis choices on figure 3 seem opposite your written story. Typically the x-axis is more causal (the IV) and the y-axis is the DV. Of course, this data is correlational, so it could go either way, but your story is about how a negative environment (ie, a low GDP country) leads to "impatient" choices, so I think it would make sense to flip your axes, so the story is consistent. It would also make it easier to eyeball the time preference differences across conditions (in other words, we could see the anomalies more easily). To be clear, I'm not "requiring" this change as a reviewer -- just a suggestion.
- pg 4, "overpaying" should be "over paying"

Overall, great paper, should definitely be published. Good luck to the authors!

Author Rebuttal to Initial comments**REVIEWER COMMENTS:**

We have responded to all comments in an itemized, structured way, and left short notes with an indication of where to find edits in the manuscript/supplement to (hopefully) easily identify revisions. There is no doubt that this feedback has helped us to produce a better paper and we greatly appreciate the input from all reviewers. We hope you find our responses and subsequent edits to the content both sufficient and compelling.

Reviewer #1:**Remarks to the Author:**

First of all, I'd like to thank the authors for this interesting submission. The main topic of this paper - exploring how temporal discounting and its anomalies are related to economic factors and what extent these relationships are generalizable across culturally diverse populations (61 countries) - fits the scope of Nature Human Behaviour and is highly relevant for its broad readership. There are many positives to the paper. In general, the paper is well-written and its main messages, temporal discounting and its anomalies, are robust on a global level; living in a financially challenging environment is associated with higher temporal discounting - clearly expressed in the main text as well as in the abstract. The theoretical background is succinct and cites most of the literature relevant to the topic. The data collection procedure (including the a priori simulations available at the OSF), as well as its processing, were thoroughly designed. This is nicely documented in the supplementary materials and in the pre-registration. The analytic workflow is to a very high level and, as far as I was able to understand the calculations (I'm not an expert in GAM), I didn't notice any major mistakes while reviewing the functions/code. However, I wasn't able to properly run the code and reproduce the results but I'll get to that point later. The study was pre-registered and the authors transparently disclose the deviations from the pre-registration. Despite the complexity of the analyses, the results are comprehensively written and visualized, and the interpretations (usually) logically follow the findings. All in all, I'm very pleased with the paper and I think that the authors did a great job. The findings and their robustness (not only the methodological/statistical rigor but also the cultural diversity of the sample) significantly advance the field of temporal discounting and its association with economic circumstances. Below, I provide some (mostly minor) comments and suggestions on how to improve the paper and supplementary materials.

Suffice to say we are extremely grateful to the reviewer for these encouraging and detailed comments. We have responded to every comment with the intention of resolving in-full, and have used numbers to

make it efficient to find where the edits have been made in the manuscript. This feedback has been extremely valuable and helped us to produce a better manuscript, so we want to extend our appreciation to the reviewer for being so diligent in this review, and also for the positive views about the work in general. We hope comments and edits to the manuscript, code, and supplement are sufficient, but of course we welcome any further suggestions following this revision.

Theoretical background and research questions

The theoretical background flows well and is succinct. The strengths and novelty of the authors' approach to addressing these problems are nicely summarized. However, I believe it would benefit from some restructuring in the main text and additional information in the supplementary materials.

(1) In particular, I'd welcome a clearer paragraph on the aims of the research. While they are in the text, the information is scattered across three paragraphs and also blends in with the paragraph about inflation. A summary of these aims would help the reader grasp the idea of the paper right from the beginning. Maybe moving the information about assessing temporal discounting from the first paragraph to the beginning of the third paragraph on page 3 would be enough although perhaps formulating a paragraph that mentions time discounting, its anomalies and potential economic determinants would be a better choice.

Thank you for bringing this to our attention. We have now made more explicit the precise aims in the first paragraph after Figure 1 and have also made the order switch as suggested between those two paragraphs.

Edit location: Page 2, 3

(2) It would also help to briefly mention the anomalies earlier in the text – for example, in the first paragraph on page 2.

We have now introduced anomalies in the opening paragraph.

Edit location: Page 2 (multiple places)

(3) The authors write that "This tendency, known as temporal discounting, is routinely associated with lower wealth" and support the statement with several references. Could the authors provide a brief, quantitative, synthesis of these (and maybe also other findings) on this topic as part of the supplementary materials? This is not necessary for the paper per se but it would be very informative to summarize the available evidence (maybe this would also help in setting informative priors in the Bayesian analyses if the authors or an interested reader decided to do further sensitivity analyses, although I guess it wouldn't change the results in this case).

We fully agree with this point and invested considerable time into providing a detailed insight. However, if we understand them correctly, journal guidelines prohibit using the supplement for extending text not related to methods or results. As these reviews will be published as a supplement of their own with the manuscript (if accepted), we hope it is alright that we simply provide the response here:

- 5 studies found a negative relationship between income and discounting ^(2, 3, 4, 8, 10)
- 1 study found a negative relationship between wealth and discounting ⁽⁷⁾
- 2 studies found that lower income older adults demonstrated more extreme discounting than upper income older adults and upper income younger adults ^(1, 9)
- 1 study found that scarcity leads individuals to engage in certain problems while neglecting others, resulting in behaviors including overborrowing ⁽⁵⁾
- 1 study reports a positive relationship between socioeconomic status, inhibitory control and self-regulation ⁽⁶⁾
- Carvalho et al., 2016 results do not support the claim that that financial strain by itself worsens the quality of decision-making. However, further indicating that scarce resources indeed can affect one's willingness to delay gratification ⁽¹¹⁾

⁽¹⁾Green, L., Myerson, J., Lichtman, D., Rosen, S. & Fry, A. Temporal discounting in choice between delayed rewards: The role of age and income. *Psychology and Aging* **11**, 79 (1996).

⁽²⁾Falk, A. *et al.* Global Evidence on Economic Preferences. *The Quarterly Journal of Economics* **133**, 1645–1692 (2018).

⁽³⁾Adamkovič, M. & Martončík, M. A Review of Consequences of Poverty on Economic Decision-Making: A Hypothesized Model of a Cognitive Mechanism. *Front Psychol* **8**, 1784 (2017).

⁽⁴⁾Brown, J. R., Ivković, Z. & Weisbenner, S. Empirical determinants of intertemporal choice. *Journal of Financial Economics* **116**, 473–486 (2015).

⁽⁵⁾Shah, A. K., Mullainathan, S. & Shafir, E. Some consequences of having too little. *Science* **338**, 682–685 (2012).

⁽⁶⁾Sheehy-Skeffington, J. & Rea, J. *How poverty affects people's decision-making processes*. 79 <https://www.jrf.org.uk/report/how-poverty-affects-peoples-decision-making-processes> (2017).

⁽⁷⁾Epper, T. *et al.* Time Discounting and Wealth Inequality. *American Economic Review* **110**, 1177–1205 (2020).

⁽⁸⁾Lawrance, E. C. Poverty and the Rate of Time Preference: Evidence from Panel Data. *Journal of Political Economy* **99**, 54–77 (1991).

⁽⁹⁾Burro, G., McDonald, R., Read, D., & Taj, U. (2021, December 20). *Patience decreases with age for the poor but not for the rich: An international comparison*. *Journal of Economic Behavior & Organization*.

⁽¹⁰⁾Ludwig, R. M., Flournoy, J. C., & Berkman, E. T. (2019). Inequality in personality and temporal discounting across socioeconomic status? Assessing the evidence. *Journal of research in personality*, *81*, 79-87.

⁽¹¹⁾Carvalho, L. S., Meier, S., & Wang, S. W. (2016). Poverty and Economic Decision-Making: Evidence from Changes in Financial Resources at Payday. *The American economic review*, *106*(2), 260–284. <https://doi.org/10.1257/aer.20140481>

Methods

As I wrote earlier, the data collection was well planned, and the research methodology was thoroughly designed. In order to assess statistical power, the authors ran a priori simulations for both GLM and

HLM models. Most of the details are sufficiently explained in the supplementary materials. The authors follow the pre-registration (even though the pre-registered hypotheses were somewhat vague - resembling research questions rather than testable hypotheses) and transparently report the deviations. I still, however, have some questions/suggestions.

As stated before, we are tremendously grateful to the reviewer for these kind and encouraging comments. All specific requests are addressed in order, again with numbers that should assist finding where they were changed in the main manuscript.

(4) (1) Could the authors clarify how the total temporal discounting score was computed? I understand that the authors combined all items (“...we computed a discounting score based on responses to all choice items and anomalies, which ranged from 0 to 19...”). However, I’m not convinced that such an approach delivers a reliable/valid score. Could the authors provide a rationale (and, ideally, include it in the supplementary materials) regarding the reliability, but mainly validity of this score?

This is a very important point and we have now added a lengthy statement in the supplementary methods that covers reliability and validity of the scoring approach as well as a reiteration of the score composition itself. We hope this goes beyond the request to cover additional points for posterity and robustness, but please let us know if anything needs to be clarified or elaborated further.

Edit location: Supplement Page 4, 5

(5) (2) Furthermore, the authors did a nice job regarding the data cleaning and set rational inclusion criteria. Some of their choices (e.g., removing participants who had completed the survey in less than 120 seconds; removing “extreme outliers larger than 100 times the median absolute deviation above the country median for income and 1000 times larger than the median absolute deviation for national median assets”) could be considered too liberal. I wonder to what extent the results would change if more conservative inclusion criteria were applied? Could the authors run a sensitivity analysis on this?

First, we appreciate the acknowledgement of the data management - and we especially appreciate the time that the reviewer put into checking all aspects. We have conducted some basic sensitivity analyses in our main models. In these tests, we considered only individuals with incomes up to 25 times the median absolute deviation and with assets up to 250 times the median absolute deviation. This new dataset included only 13100 individuals, with a maximum income of \$875,916 (.20 times the original one), and a maximum assets of \$21,167,970 (.41 times the original one). Results presented in Table S17 showed that our initial and main results were largely unchanged, with the only exceptions being the non-linear effects of economic inequality becoming insignificant for present bias and absolute magnitude. However, in our final models, we already considered the effect of inequality to be linear, so our main results are left unchanged. Therefore, our view was that the initial results were robust to the

income/assets exclusion criteria but welcome further input on this as inclusion/exclusion was a major topic of consideration given the nature of the sample.

(6) (3) Besides the delay-speedup scenario, was any randomization concerning the measures/items applied?

Due to the need for certain items to appear earlier in order to set the anomaly values, most items had to follow a specific order. We did consider randomizing present bias and subadditivity along with the delay-speedup scenarios. However, this created some concerns when code was being checked because all values were contingent on earlier choices and small issues appeared inconsistently between different country collection apparatuses. As this only related to three measures, all of which had similar values and natures, and because a similar method used in work related (reference below) to this showed no order effects, we only randomized the delay-speedup scenarios. Some text has been added in the methods about this.

Reference: Supplement Figure F2 in Ruggeri, K., Alí, S., Berge, M. L., Bertoldo, G., Bjørndal, L. D., Cortijos-Bernabeu, A., ... & Folke, T. (2020). Replicating patterns of prospect theory for decision under risk. *Nature Human Behaviour*, 4(6), 622-633.

Edit location: Page 15

(7) (4) The authors tested temporal discounting and its anomalies across different scenarios, which is (as noticed also by the authors themselves) a very strong point of this paper. As is typical for research on temporal discounting, researchers' degrees of freedom play a huge role in creating the items (e.g., the usage of 10% of monthly household income; delayed values of, say, 150%; waiting period of one year, etc.). Could the authors discuss (in the supplementary materials) why they made these choices and how researchers' degrees of freedom could potentially affect the results in this case? I know this might be a difficult request, so please just consider it as a suggestion to strengthen the robustness of the study.

Actually, we are happy to answer this and have added this explanation to precede the opening of the variable definitions in the supplementary methods as it seems most appropriate there. It is a great point and hope the response clarifies. In short, we did go through many iterations and discussions of different approaches, but our primary bases were recent experience (in the same study reference in the prior response), pragmatics, participant engagement, and comparability to related research. Because we wanted to ensure a large sample across countries, particularly those often not included in such studies, we wanted to keep the items limited, easy-to-understand, and not appear to 'trick' anyone with complex values that might have intimidated some sort of optimal-suboptimal continuum. Using the base values chosen, this meant mostly very simple value comparisons (e.g., 500 vs 550) and not one 'clean' (e.g., 500) and one peculiar (e.g., 483.33), while keeping the exact ratio the same across all countries.

We also did not want this shortened version to be too distinct from similar studies so as to limit what inferred validity we would have from similar methods. These values were simple, consistently understood (noting not all countries use same calendar, though), and made use of values large enough that comparative differences between values/PPP by country using the estimates to produce 'clean' numbers would not substantively impact their perceived value. In other words, some values were rounded to slightly above the average and others, slightly below. However, this was done randomly and at no point was it substantively different in a way that would affect interpretation.

Edit location: Supplement Page 3

(8) (5) The authors have used an interesting method of participant recruitment. Since this is a global study with a potentially significant impact on the field and the related generalizability of the results, could the authors provide a supplementary table which would compare how the recruited sample differs from the population of each country in basic demographic characteristics (i.e., gender, age, completed education; of course, if such data are available for the countries)?

This is a very important point and we also invested heavily in a comprehensive response. There is now a full table with these demographics (though only at the level available, some aspects are not precisely as we collected them). We have also made a mention of this in limitations.

Edit location: Supplement Page 26, 43-44; Manuscript page 6

One more thing - to preserve the participants' anonymity, please remove variables like IP address or location from the raw dataset.

All data are completely anonymized and IP addresses deleted from all versions of data files.

Results

The authors nicely extract and report the most relevant findings and enrich them with additional results in the supplementary materials. However, as there are so many analyses, it was occasionally hard to follow them in all documents (main paper, supplementary material, code) so I might be wrong in some of my following comments. However, I've noticed several things I'd like to share with the authors.

Thank you again and we acknowledge there were many analyses included. We did attempt to keep these consolidated and systematic, but appreciate it was a lot to review as diligently as you have. We hope our

responses clarify any concerns, and we have made a number of edits in the materials in response to them. Please let us know if any are not sufficient to the concerns raised.

(10) (1) Marginal differences in p-values or ORs lead to substantial differences in the interpretations. For instance, the authors write that they found "...a positive effect (of GDP) for delay-speedup (OR = 0.95, 95% CI [0.91, 0.99], p = 0.002)" and in the next sentence they state "...we found no effect on the remaining anomalies (0.95 < OR < 1.01; 0.027 < p < 0.688)". I understand that this is based on p-values/CIs containing zero (and probably also on BFs), but the ORs are not really different.

We acknowledge this point and have added a clarification statement at this precise location in the manuscript. It may also help to iterate that we intentionally used "<" and ">" in the final parentheses as we were highlighting the endpoints and not the actual ORs/ps. We hope the added text clarifies but we can also add more if felt necessary.

Edit location: Page 8

(11) (2) On the other hand, on one occasion, the authors state they observed „...only modest variability (of temporal discounting) for national means“. I'm not sure if this is true - both relative (I2) and absolute (τ^2) values of heterogeneity are (relatively) high. This can be nicely seen in the example provided by the authors who compared Japan (M = 8.1, SD = 3.9) and Argentina (M = 15.1, SD = 3), with the difference for these two countries being Cohen's $d \approx 2$. Speaking of high heterogeneity, maybe the authors could try running a meta-regression to examine its source.

We agree that this is a very important aspect to be clear on and are thankful for the detailed review of the text. As we mention in the text, differences between country-specific means were as high as 20% (for scores) in the best-case scenario (i.e., a null model only including a country random intercept). More than half the variance at country-level variation is explained in our final models, including country-level predictors such as GINI or GDP (ICC = 0.10, Table S8; ICC = 0.07; Table S10). As such, it is our assessment that while we observed a non-negligible variability at country-level, the same plays a lesser role when compared to within-country and residual variance. For the anomalies, variance at country-level is substantially lower than in the case of scores when considering a null model with a random intercept (between 2-8%), so the differences between countries are even lower. Be that as it may, we agree with the reviewer that the specific role of predictors should be explored in the future, where meta-regressions and other complex effects (exploring random-slopes) should be inspected in detail and we would like to make it a focal point of future analyses we run and/or recommend with these data.

(12) (4) Following this, the authors conclude that "this is largely based on finding remarkable consistency and robustness in patterns of intertemporal choice across 61 countries, with substantially more variability within each country than between their means.". I'd suggest a slight reformulation of this statement although maybe I just missed something, and the conclusion is correct.

Similar to the point above, this is really a critical point and we are happy to clarify. In our view, the country with the lowest variability is Pakistan, which a range of 12 points (6 to 18). Concurrently, three countries (Indonesia, Nigeria, and Romania) had the maximum range of 19. The range from highest to lowest mean score between countries is 7.0 (Japan = 7.1 to Argentina = 14.1). We certainly do not mean to imply there is no variability between countries nor that the variability is not meaningful. Our only point is that, as much as between-country comparisons are popular (e.g., “which country is least patient? most patient?”), the variability within is at least as important as the overall variability, and certainly more indicative of just comparing country means. We hope this clarifies our point but are happy to provide further discussion or analysis if it would be useful. We considered just removing “substantially” from the text but hope this explanation would be sufficient to address the concerns. If you feel that may be beneficial, we can remove it.

Statistical analysis, code, and data

The analytic workflow described in the paper/supplementary material is precise, and the proposed statistical analyses are appropriately used and technically sound (at least to the extent I was able to understand them), thus allowing for robust findings. However, I struggled while reviewing the code / trying to reproduce the findings. The code is not fully reproducible (I’ve tried it on two different machines as I was expecting some potential problems with RStan) and would need some adjustments and more detailed annotation.

We are tremendously thankful for you putting such a thorough review into the article and specifically into the code. As you will see, we have now addressed all of your comments and resolved any requests you have made for improvements in the code itself. We also want to highlight that, in trying to understand why the code was not working on other machines, we found the code for scoring was inadvertently adding 1 to each participant. This has now also been resolved (though it only affected some descriptives and shifted on visual; nothing else was impacted due to the use of standardized values in most analyses). We are also running the updated code on a number of different machines to make sure it works for everyone, and will be confident this is the case before uploading the final on psyarxiv.

(13) (1) First of all, I’d like the authors to check the code (push-button style) on different machines.

This has been done and we will continue to check that the code works on multiple computers prior to making public on OSF.

(14) (2) Please include the output of sessionInfo() in the script.

This has been done.

(15) (3) I was able to run file 2_1 with the sourced data (although with several errors) but not with data produced by file 1_1. The same applies to the graphs (3_1) and supplementary analyses (4_1). Besides the unreproducible parts, I've noticed two substantial typos: the function fit_models_lmer contains a mistake in line 571: mg5 = mg6 should be mg6 = mg6; file 2_1, line 101 should be cmdstanr instead of cdmrstan.

We apologize for whatever issue occurred with the shared code and have now resolved this. Furthermore, to clarify our changes, the functions declared in lines 571 were deprecated and removed from the code. Thank you for bringing this to our attention.

(16) (4) More detailed annotation would be very much welcomed to understand better what's happening in the code. I'm aware that preparing a more annotated code is a difficult and unpleasant job. However, it allows for easier comprehension and easier reproducibility by independent researchers, which I find extremely important.

We understand this and had attempted to do much of this in the original version. We hope the revised version helps even further but please let us know if any additional aspects or annotations would be useful.

(17) (4) Please consider using standard scripts instead of RMarkdown files, although this is just a matter of preference.

This has now been done.

Discussion/conclusion

The discussion is concise and summarizes the main findings and contributions of the study. Although I enjoy reading short yet succinct introductions and discussions, I believe that some adjustments could strengthen the study even more.

We are very pleased at this comment and have attempted to address each point raised systematically. Thank you again.

(18) (1) I'm missing a paragraph about the limitations of the study (e.g., naturally, there's a difference between hypothetical and real scenarios, although evidence on the correlation of the two differs) and perhaps also a few additional sentences on the robustness of the study.

We regret this omission and appreciate it being brought to our attention. There is now a comprehensive response in the end of the results section covering these points and others (some raised by you; some raised by other reviewers).

Edit location: Page 6

(19) (2) The authors also depict some possible implications (e.g., for policymaking) but don't elaborate on that any further. Covering these issues in more depth (not necessarily in the main text – these suggestions could be part of the supplementary materials) could make the paper even more robust and comprehensive.

We appreciate the invitation to elaborate on this, though we were intentionally a bit cautious about drawing a straight line from one study to specific policies. That said, we've added a full paragraph in the conclusion about specific policies that may find these results relevant, though we stop short of suggesting they should automatically be revised (or how).

Edit location: Pages 10-11

Very minor comments

(20) (1) In the introduction, the authors write "To test our hypotheses...". Besides one exception in the paragraph about the effects of inflation on temporal discounting, the paper doesn't contain any hypotheses (instead, it contains goals of the research), so reformulation would help.

We have now made clear the set of hypotheses in the end of the introduction and also made a more explicit link to the pre-registration, where the full plan is detailed.

Edit location: Page 3

(21) (2) In the first sentence of the last paragraph on page 4, please add that it was based on average income per month.

Confirming this has been done.

Edit location: Page 5

(22) (3) The final dataset (the dataset on which the analyses were performed; dat_filter) consists of a sample of 13,324 (becomes 13,373 when I run file 1_1 instead of loading 1_2_data_files.RData provided by the authors) instead of the 13,629 participants the authors mention in the text.

For posterity, we wanted to clarify the differences, though there is no actual discrepancy. The 13,629 participants refer to the complete dataset after our pre-registered data exclusions. For our final analyses, we included two additional exclusions (unrealistic average income and assets responses, and whether individuals reported to have no income but be employed full or part-time), which removed 305 individuals, and left our final sample size to be 13,324. We report these as such to maximize transparency about the process, and because our original criteria (along with the attention check) is how we identified who qualified as having completed the instrument, with the exclusion done for analyses. If

this is still seen as problematic, we can simply refer to 13,324 throughout, with this approach mentioned in the sample descriptions.

My final remarks

This study has the potential to be one of the most influential papers on the topic of temporal discounting. Once again, I'd like to thank the authors for this submission, and I hope they'll find some of my comments and suggestions useful.

**All the best,
Matúš Adamkovič**

This was an extremely encouraging statement to read and given how detail-oriented this review was, we are especially pleased by it. We hope you find the edits made to be sufficiently in line with these comments but obviously welcome further feedback if you feel additional improvements are warranted.

Reviewer #2:

Remarks to the Author:

I am positive towards this work. I believe that the dataset presented in the paper is extraordinary and that a great effort was devoted to produce this piece of research. I also believe that the main conclusion that temporal discounting is a global phenomenon, is very relevant to the literature. I have just a few comments on this work:

We were delighted to read this general assessment as well as the individual comments. They have been extremely helpful and we have addressed each of them systematically where requested, plus provided brief responses to each point. Please do let us know if you feel further edits are warranted or if we have missed any of the critical points.

(23) 1) The abstract is very focused on the finding that economic inequality is a predictor of time preferences. However, I am not sure this is the main message which you get from the paper. Plus, economic inequality is correlated to so many other outcomes. I would tone down all those places where you attribute your finding to economic inequality and clearly state it is very correlational.

We acknowledge these points and agree with the premise. To resolve this, we have adjusted the abstract (noting some of the language is limited by word counts) and the final statement clarified as correlational.

Parts in the results have been modified as well, just to avoid any implications of causation. You may spot a number of minor changes in other sections to account for this correction.

Edit location: Manuscript Page 1, 8

(24) 2) The structure of the introduction is not so clear. Some findings are highly emphasized (like inflation) at a point where I would have expected the focus to be on other points. Plus, I am not sure that the spending pattern of the CARES Act is a necessary justification for your work. Most importantly, you give the impression that you are going to focus a lot on individual financial conditions, which is not so true in the rest of the paper.

As we certainly want clarity on each of these points, we did something of a minor overhaul of the introduction section, which also aligned with some of the comments of other reviewers. First, rather than minimize the point on inflation, we instead attempted to position each aspect with similar depth and followed an order of relevance, making sure each of our primary aims/hypotheses had a clear explanation for context. We also attempted to make sure the introduction text (especially the first paragraph and someafter Figure 1) establishes better the context for why CARES Act patterns were relevant. Finally, in that section and throughout the paper, we have been less relaxed about saying “individuals” or “groups”, and adjusted wherever appropriate (note: in a few instances, this came down to avoiding over-use of words such as “people”, “participants”, “groups”, and “individuals”, and though we may have intended them generically in some cases, we recognize how that may create implications for reading). We hope those edits have resolved these concerns but please point us to any remaining issues with these terms if you have any.

Edit location: Page 2

(25) 3) In the conclusion, you highlight that you find consistency and robustness of the intertemporal anomalies. I think this is the most robust point and the main innovation of your work.

We really appreciate this comment and have highlighted it in the abstract and introduction as well.

Edit location: Page 1, 3, 6

(26) 4) To sum up, it is not clear what your main focus is. I would say the point in 3, since the conclusion section seems quite clear and structured. I invite you to harmonize abstract/introduction and conclusion. Plus, I think it would be beneficial having a clear roadmap of the hypotheses that you want to test and a table which summarizes which hypotheses are supported by the data. It is quite difficult getting the main messages out of your work.

With the entire introduction now reformulated, and various changes throughout with an emphasis on stating aims/hypotheses outright, we hope these points are now clear. Please let us know if you think anything further is necessary to streamline/harmonize, though.

Edit location: Page 2 (multiple places), 3

(27) 5) I think that that this work is too premature to claim that “It also disrupts any notion that lower-income individuals are somehow concerning decision-makers, as negative environments are widely influential.” There are not many analyses referring to individual financial conditions. Plus, when you refer to individual financial conditions, I would use a relative measure of wealth and not an absolute one like assets. I suggest you show how individuals ranking at a given percentile of the income distribution (for example) behave, depending on how unequal their country of origin is. Until that point, I think your conclusion is premature and you should tone it down.

To the first point, we have softened that statement somewhat as we agree it may have been overwrought. As for the ‘individuals’ point, this should now be resolved based on the earlier response, but to reiterate, we have made adjustments throughout and hope that all parts are either explicit what they relate to or are clearly meant as generic terms for the sample.

To the second point about income percentiles, we have produced the following visual to hopefully relieve this concern. Here we plot the relationship between the income quintile (in the overall distribution of incomes) and scores for four groups of countries (based on Gini). There is a minor effect (see the axis going from 5 to 12). Basically, individuals with low to average income (10th to 40th percentiles) present higher scores if and only if belonging to a high-inequality country (purple and green lines). In more equal countries, income does not have a large influence on scores. Interestingly, high income individuals (80th or 90th percentile on the world income distribution) from highly unequal countries are expected to have higher temporal discounting than someone poor (on the opposite side of the income distribution; 10th or 20th percentile) from a highly equal country. Though not perfectly descriptive, we feel this supports our view that negative environments are highly influential (even if it is a small effect), and even wealthy people on highly unequal countries would be show a great rate of temporal discounting.

Edit location: Page 10

(28) 6) Inflation has much more focus in the introduction than in the rest of the paper. Plus, do you have reasons to expect a difference due to only very high inflation?

As we noted earlier related to the earlier comment, our modifications should now temper this and the text should be roughly balanced between the various hypotheses and topics we analyze. We also adjusted some wording in that paragraph as it indeed implied that inflation overrides all, which is not our view. Please let us know if you feel this needs further correction.

Edit location: Page 3 (multiple places)

(29) 7) There are several studies which measured time preferences at an almost global level. You only refer to Falk et al. (2018). It would be interesting to see a comparison of your data with all these other data. However, I realize that your data are more detailed and that they will probably be the gold standard for the future. Other papers include: Burro et al. (2022) Patience Decreases With Age for the Poor but Not for the Rich: An International Comparison (JEBO); Wang et al. (2016) How time preferences differ: Evidence from 53 countries (JOEP); Rieger et al. (2021) Universal time preference (Plos One). In particular, in the last paper (the Plos One), the authors list a series of global measures of time preferences and they aggregate them into a single one.

We greatly appreciate this list of recommended references and have reviewed each, incorporating into the introduction section. Based on what we understand to be permitted in the journal, we hope the way

we compare the data is what you had in mind. We considered extracting more and putting into a table in the supplement but think this may not be permitted in the journal (as it extends text rather than complements methods/results). With that said, we feel this was a really valuable inclusion in the introduction as it is and welcome any further comments.

Edit location: Page 3

(30) 8) Some sentences seem incomplete (description of fig 4 for example) and I think that there are a few typos (“evaluated a variety of Bayesian approaches” for example).

Thank you for pointing these out. We have corrected both.

Edit location: Page 25

Reviewer #3:

Remarks to the Author:

In this paper, the authors present a very large dataset examining intertemporal choice phenomena across 61 countries and over 90 participants in almost every country studied. By analyzing the choice proportions across several scenarios, designed to elicit overall patterns of discounting as well as several important anomalies of intertemporal choice, they show that these anomalies largely appear consistently across countries, but that these anomalies – and especially overall patterns of discounting – vary according to both individual and country-level income as well as indices related to inequality (GINI), inflation, and wealth / assets.

My immediate impression is that this is among the largest and arguably most important data sets ever collected on intertemporal choice. The paper is clearly written and provides a straightforward set of analyses that highlight the main findings. Most of the issues and recommendations I have are relatively minor and aimed at making sure that the manuscript is as clear as possible and reaches a wide audience.

We were tremendously encouraged by these extremely positive remarks and took each comment you made very seriously to ensure we met the standard you gave. We hope you find our responses to these issues sufficient but of course welcome any further suggestions to resolve the topics you point out. We also hope you feel the points we make in trying to guide others toward potentially meaningful re-analysis of our data have been made well. We look forward to any further feedback you have and are again thankful for the positive words.

(31) 1. The biggest question I had while reading was: What was the attention check that was added to the study? This was a deviation from the pre-registration, as the authors note, but I don't see anywhere that the nature of the attention check was discussed. I know that sometimes in intertemporal choice work a "dominated" option is offered (e.g., \$100 now or \$50 in a year), and it could be the case that the prevalence of this type of responding across countries is theoretically interesting. But I could not find the attention check items in the supplement or anywhere in the paper, so it is hard to say whether they would be interesting or not. Given that over 6000 participants failed the attention check, it seems like this could be an important issue (as well as its translation across languages).

Thank you for pointing this out. We have now added in a brief statement in the method to clarify the attention check. We used something similar to what you describe, but made it even more evident as it was a "gain now or lose later" scenario. Our view was that this was the simplest way to ensure understanding of the questions and paying attention, though we recognize someone not paying at all might have been equally likely to have chosen the correct option, which we aimed to address with the timing/speed exclusion. We also wanted to note that all of the materials (all languages) will be available through the OSF link (it should be the same one already in the manuscript) in case anyone would like to re-attempt with a different version. That said, please let us know if you feel anything further is necessary.

Edit location: Page 11

(32) 2. Another issue that is likely to come up with some readers is the fact that participants were not compensated for their participation (except in the Japanese sample?), which may be taken by many (especially economists) to mean that participants were not motivated to exhibit rational / optimal behavior. It may be worth heading off this criticism in the paper. Here are some papers that may help in doing so:

Wiseman, D. B., & Levin, I. P. (1996). Comparing risky decision making under conditions of real and hypothetical consequences. *Organizational behavior and human decision processes*, 66(3), 241-250.

Kühberger, A., Schulte-Mecklenbeck, M., & Perner, J. (2002). Framing decisions: Hypothetical and real. *Organizational Behavior and Human Decision Processes*, 89(2), 1162-1175.

Amlung, M., & MacKillop, J. (2015). Further evidence of close correspondence for alcohol demand decision making for hypothetical and incentivized rewards. *Behavioural processes*, 113, 187-191.

Madden, G. J., Begotka, A. M., Raiff, B. R., & Kastern, L. L. (2003). Delay discounting of real and hypothetical rewards. *Experimental and clinical psychopharmacology*, 11(2), 139.

Locey, M. L., Jones, B. A., & Rachlin, H. (2011). Real and hypothetical rewards. *Judgment and Decision making*, 6(6), 552.

We really appreciate this point and fully agree that it is worth explaining in more detail. Using these references, we have discussed this in the limitations sections along with others raised by the reviewers. That said, we found your list of references to be exceptionally helpful and have included all of them as suggested, though we welcome any further suggestions on this section if you feel more is warranted.

Edit location: Page 6

(33) 3. It has also become more common in intertemporal choice literature to look at not just indifference points but generative model parameters related to discounting. There are a few reasons for this, including the observation that these parameters provide a more reliable and (at least predictively) valid account of behavior on intertemporal choice tasks. This can naturally be improved with methods like hierarchical Bayesian estimation of discounting rates (k) and/or more sophisticated models that can predict all of the anomalies that are mentioned in the paper (e.g., direct difference model; Dai Busemeyer, 2014).

Odum, A. L. (2011). Delay discounting: I'm ak, you're ak. *Journal of the Experimental Analysis of Behavior*, 96(3), 427-439.

Dai, J., & Busemeyer, J. R. (2014). A probabilistic, dynamic, and attribute-wise model of intertemporal choice. *Journal of Experimental Psychology: General*, 143(4), 1489.

Molloy, M. F., Romeu, R. J., Kvam, P. D., Finn, P. R., Busemeyer, J., & Turner, B. M. (2020). Hierarchies improve individual assessment of temporal discounting behavior. *Decision*, 7(3), 212.

I am not suggesting that the authors need to carry out sophisticated model-based analyses of the results (although this would certainly be interesting!), but it is worth mentioning alternative approaches to data analysis on intertemporal choice tasks that might elucidate other interesting individual or country-level differences in behavior. This could even be spun as a future research direction that will be enabled by making the data set openly available, as the authors apparently plan to do.

We had an enjoyable discussion in our team about these points and would love if others might make use of our data in that way in the future. In that vein, we also agree that explicitly modeling the discount rates is beyond the scope of this article. Given the current controversies regarding the benefits and drawbacks of hyperbolic models, we would be particularly thrilled to understand how our original analyses and these elaborate alternative complements could complement each other in the future. That all said, we definitely agreed that we should provide some sort of road map for those that might be interested in doing so and have added a text with some of the suggested references in the supplement. For simplicity, that text is below:

There are multiple approaches to 'scoring' temporal discounting, and decomposing the concept broadly into subcomponents. In our case, we favored an atheoretical, simple approach to estimate effects. However, intertemporal discount rates have could also be explicitly modeled using novel developments that go beyond estimating discount rates employing an hyperbolic model^{2,3}. Recent alternatives based on hierarchical models employing Bayesian estimation have been shown to be particularly informative^{4,5} and should be considered for future exploration with ours or other data.

Edit location: Supplement Page 5

Reviewer #4:

Remarks to the Author:

This is a great paper. The design and analysis choices are excellent and follow the current standard. For example, I'm happy that they report the raw number of SS vs LL choices (reversed for losses), rather than calculating discount rates with a hyperbolic model or similar.

We are delighted to hear this and extremely thankful for the feedback as well as the individual comments. To the extent possible, we have attempted to address all concerns raised in detail, both in the replies and in the study material. You will see our one note in the second comment, but otherwise, hope you find the rest of our responses to be sufficient. Any further feedback on this would be welcomed.

Here are my suggestions for improvement:

(34) 1. - I'm not sure what the policy is at this journal, but it would be great if you would publicly share the (deidentified) data and the data analysis scripts from this project, via something like <http://osf.io> .

As you point out, there has been a replication crisis and crisis of confidence in behavioral sciences, and we need transition to open science standards.

We absolutely agree and as now indicated in the manuscript more clearly, we have already established the location for the files on psyarxiv, and all (deidentified) data will be made available soon after publication (if accepted).

(35) 2. - although I love the # of SS choices for the primary analyses and reporting, it would also be interesting and useful to see them converted to annualized discount rates (using the continuously compounded exponential model) for comparison with other studies and economic conditions. A table that shows the annualized discount rate as a percentage in each country and each condition would be nice, similar to the Loewenstein & Prelec (1992) paper.

We are very pleased with this and certainly recognize the points as critical but had a slight issue resolving them. Forgive us if we are mistaken, but we believe you were referring to Table 1 in the Loewenstein & Prelec (1997) paper. If so, we actually had already planned to provide something like this, but with all of the additional country detail, in the psyarxiv files, and not including the exponential aspect given our approach and findings. The material we put together is a very large repository of all the values used and their reference to the 'base' within each country, plus with references for how values were computed. It is quite difficult to make this file fit into an academic publication (even supplement), so we hope it is sufficient to include this screenshot as an indication that would satisfy the request. It does not look precisely the same as the table from Loewenstein & Prelec, but this is mainly because we have many countries for each value, even though the base and subsequent values maintain the same rate/relationship. It also directly fits the approach we used, so we hope this is considered a sufficient response.

A	B	C	D	E	F	G	H	I	J	K	L	M	N	O
Country	Pod	Language	Currency	Monthly household income	Leverage type (Mean or Median)	Individual or household	Year	Official reference	Conversion value base (200)	Conversion value base (2000)	Value change 1 (101% of first conversion value)	Value change 2 (102% of first conversion value)	Value change 3 (103% of first conversion value)	
USA/BASE	Anglophone	English	US dollars	5725	Median	Household	2018	https://www.ceritas.gov/india	500	5000	505	510	515	
Angola	Lusophone	Portuguese	Kwanza	15454	Mean	Individual	2019	https://www.statista.com/statistics/1096800/average-household-income-in-angola/	1500	15000	1515	1530	1545	
Argentina	Lusophone	Spanish	Argentine peso (ARS)	75390	Mean	Household	2021	https://www.index.mobi/argentina	8000	80000	8160	8320	8480	
Australia	Anglophone	English	AUS dollars	10179	Median	Household	2020	https://www.abs.gov.au/statistics/tables/chart/tables/2020	1000	10000	1010	1020	1030	
Austria	Central Europ	German	Euro	3290	Median	Household	2020	https://www.statista.com/statistics/1096800/average-household-income-in-austria/	300	3000	303	306	309	
Belgium	Central Europ	Dutch	Euro	1972	Median	Household	2018	https://www.statista.com/statistics/1096800/average-household-income-in-belgium/	200	2000	202	204	206	
Bosnia and Herzegovina	Balkans	Bosnian	The Bosnia and Herzegovina	698	Mean	Individual	2020	https://libras.uns.ac.rs/portal/	100	1000	101	102	103	
Brazil	Lusophone	Portuguese	Real (BRL)	2795	Mean	Individual	2020	https://data.bps.go.id/series	500	5000	505	510	515	
Bulgaria	Central Europ	Bulgarian	Bulgarian Lev (BGN)	1306	Mean	Household	2021	https://www.nsi.bg/znani/	1000	10000	101	102	103	
Canada	Anglophone	English	Canadian Dollar	5241	Median	Household	2019	https://www150.statcan.gc.ca/n1/pub/28-263-x/2019001/article/00001-eng.htm	500	5000	505	510	515	
China	Asia	Chinese (Simplified)	Renminbi (RMB)	2246	Median	Individual	2020	https://www.nhantriviet.com/daily/world/2020/07/2020-07-20-01	200	2000	202	204	206	
Croatia	Balkans	Croatian	Kuna	6882	Median	Individual	2021	https://www.dzs.hr/DZS/2021_01/	500	5000	505	510	515	
Czech Republic	Central Europ	Czech	Czech koruna (CZK)	17483	Mean	Individual	2019	https://www.czso.cz/eng/tables/tables.asp	2000	20000	2040	2080	2120	
Denmark	Scandinavia	Danish	Danish krone (DKK)	28274	Mean	Household	2019	https://www.statbank.dk/	4000	40000	4040	4080	4120	
Egypt	MENA	Arabic	Egyptian Pound (EGP)	4904	Mean	Household	2018	https://www.ceritas.gov/india	500	5000	505	510	515	
Estonia	Balkans	Estonian	Euro	815	Mean	Individual	2019	https://andromeda.ee/andromeda/	100	1000	101	102	103	
Ethiopia	Balkans	Amharic	Ethiopian Birr (ETB)	12962	Mean	Household	2020	https://www.ethiopia.gov.et/	1000	10000	1010	1020	1030	
France	Central Europ	French	Euro	1919	Median	Household	2019	https://www.insee.fr/fr/statistiques	200	2000	202	204	206	
Georgia	Eurasia	Georgian	Georgian Lari (GEL)	1175	Mean	Household	2019	https://www.geostatistics.gov.ge/	100	1000	101	102	103	
Germany	Central Europ	German	Euro	2738	Median	Household	2018	https://www.destatis.de/DE/Pressemitteilungen/2018/08/18-081.html	400	4000	404	408	412	
Ghana	Balkans	English	Gh Cedis	2828	Mean	Household	2017	https://www.statbank.gov.gh/	300	3000	303	306	309	
India	Asia	Hindi	Indian Rupee (₹)	12029	Mean	Individual	2020	https://www.india.gov.in/	1000	10000	1010	1020	1030	
Indonesia	Asia	Indonesian	Indonesian Rupiah (IDR)	819718	Median	Individual	2020	https://www.bps.go.id/	10000	100000	10100	10200	10300	
Iran	Eurasia	Farsi	Toman	3000000	Mean	Household	2017	https://www.irica.gov.ir/	400000	4000000	404000	408000	412000	
Ireland	Anglophone	English	Euro	3230	Median	Household	2019	https://www.csa.gov.ie/	400	4000	404	408	412	
Israel	MENA	Hebrew	Israeli New Sheqel (NIS)	21083	Mean	Household	2018	https://www.csa.gov.il/	2000	20000	2040	2080	2120	
Italy	Central Europ	Italian	Euro	2837	Mean	Household	2018	https://www.istat.it/it/	300	3000	303	306	309	
Japan	Asia	Japanese	Japanese Yen	490200	Mean	Household	2019	https://www.e-stat.go.jp/en/statlist/contents.do	50000	500000	51000	52000	53000	
Jordan	MENA	Arabic	Jordanian Dinar (JOD)	637	Mean	Household	2018	https://www.dgs.gov.jo/	1000	10000	101	102	103	
Kazakhstan	Eurasia	Russian	Tenge (₸)	210030	Mean	Household	2020	https://stat.gov.kz/	20000	200000	20400	20800	21200	
Kazakhstan	Eurasia	Kazakh	Tenge (₸)	210030	Mean	Household	2020	https://stat.gov.kz/	20000	200000	20400	20800	21200	
Korea	Balkans	Korean	Korean Shilling (KES)	10000	Mean	Individual	2019	https://www.kstat.go.kr/	10000	100000	10100	10200	10300	
Labanon	MENA	Arabic	Labanese Pounds (LBP)	1900000	Mean	Household	2019	http://www.stat.gov.lb/	200000	2000000	202000	204000	206000	
Malaysia	Asia	Malay	Malaysian Ringgit (MYR)	8000	Mean	Household	2019	https://www.bps.go.id/	800	8000	808	816	824	

(36) 3. - "anyone facing a negative financial environment – even with better incomes within that environment – is likely to make decisions that prioritize immediate clarity over future uncertainty" -- are you studying time preferences, or risk preferences? I would appreciate a deeper exploration of time, risk, and why you observe these results.

Thank you for this point and we acknowledge that statement left our aim somewhat open. We are definitely studying time more directly than risk, but we also note there is a component of risk implicit in any non-immediate prospect. Therefore, we can infer that the further away a prospect is placed in a scenario (e.g., now vs 12 months vs 2 years...), there is compounded risk+time. So if an individual has a good income now but the environment around that individual is filled with future uncertainty, the 'now' choice is not just 'now' but is also 100% certain. Contrarily, the 'future' prospect is not only 'not now', but it is compounded by delay + uncertainty about what that future looks like. We have made sure to state this outright in the same location (apologies if we went a bit too far with it) and hope this provides the discussion requested.

Edit location: Page 7

(37) 4. - A methodological limitation of the paper is that all choices are hypothetical. This makes it easier to run these study across so many different locations and teams (it's logistically very challenging to do intertemporal choices "for real"), but it does mean we have less confidence in the results, as we aren't sure if differences observed between countries and individuals are due to differences in reporting or "real" differences in time preferences. I think you should be more upfront (in the abstract, etc) about the fact that these are hypothetical choices.

This comment has been made by multiple reviewers, so we took it very seriously and have clarified in the abstract as well as elaborated in a new limitations text within the methods section. We have also added a number of references suggested by another reviewer that cover this claim and links to appropriate validation and hope those edits resolve the concern. Please let us know if more is deemed necessary to address any of the above.

Edit location: Page 1,6

(38) 5. - The axis choices on figure 3 seem opposite your written story. Typically the x-axis is more causal (the IV) and the y-axis is the DV. Of course, this data is correlational, so it could go either way, but your story is about how a negative environment (ie, a low GDP country) leads to "impatient" choices, so I think it would make sense to flip your axes, so the story is consistent. It would also make it easier to eyeball the time preference differences across conditions (in other words, we could see the anomalies more easily). To be clear, I'm not "requiring" this change as a reviewer -- just a suggestion.

We appreciate this comment and have produced the inverted visual. It does not seem to change a whole lot (two of the curves appear somewhat steeper), but we agree it may be slightly easier to read when considering the aim. Please let us know if you feel this one is better now that it is included.

Edit location: Page 25

(39) 6. - pg 4, "overpaying" should be "over paying"

Thank you, this has been fixed.

Edit location: Page 4

Overall, great paper, should definitely be published. Good luck to the authors!

Thank you again and we hope you find our edits sufficient!

Decision Letter, first revision:

Our ref: NATHUMBEHAV-211117109A

22nd March 2022

Dear Kai,

Thank you for submitting your revised manuscript "The globalizability of temporal discounting" (NATHUMBEHAV-211117109A). It has now been seen by the original referees and their comments are below. As you can see, the reviewers find that the paper has improved in revision. We will therefore be happy in principle to publish it in Nature Human Behaviour, pending minor revisions to satisfy the referees' final requests and to comply with our editorial and formatting guidelines.

We are now performing detailed checks on your paper and will send you a checklist detailing our editorial and formatting requirements within approximately one week. Please do not upload the final materials and make any revisions until you receive this additional information from us.

Sincerely,
Aisha

Aisha Bradshaw, PhD
Senior Editor
Nature Human Behaviour

Reviewer #1 (Remarks to the Author):

The authors did a wonderful job revising the manuscript and its supplementary files. The introduction (theoretical background and aims of the study) now has a better flow, it's more coherent and, importantly, easier to follow from a reader's perspective. The authors conducted further sensitivity analyses and found that their initial findings hold. The authors now discuss the main non-trivial limitations of their approach, provide a rationale for reliability and validity of their temporal discounting score, and elaborate on implications for policymaking. The SM has been enriched by an overview of general demographic profiles (Table S18), which allows for a convenient comparison of the sample used in this study with official demographic data. My comments have thus been successfully incorporated into the current version of the main manuscript and SM. I'd like to thank the authors for the additional explanations and clarifications they provided in their (honest and sufficiently detailed) responses.

I've only a few more minor comments. (1) Although I understand and (to a great extent) agree with your reasoning regarding your approach to creating a temporal discounting score, please consider supporting it (especially lines 147-177 in SM) by additional references if possible. If a similar approach hasn't been used before, you could also articulate it as an additional benefit of the study. (2) Please add more recent references to the part where you're discussing the relationship between hypothetical scenarios and real-world behaviors (main manuscript, lines 194-196). (3) I've noticed a few typos/formatting inconsistencies (e.g., main manuscript, line 28, 36; SM – "Approach to scoring").

All in all, the authors did a great job revising the paper. Thank you for your effort!

All the best,

Matúš Adamkovič

Reviewer #2 (Remarks to the Author):

I am satisfied by the authors response and I recommend acceptance.

Reviewer #3 (Remarks to the Author):

In the revision, the authors have done a good job addressing my previous comments and (as near as I can tell) those of the other reviewers. The paper is now at a stage where I think it is acceptable for publication.

I look forward to the public release of the OSF page so I can take a look at the data as a researcher rather than just as a reviewer!

Reviewer #4 (Remarks to the Author):

The authors have addressed all of the suggestions and concerns I raised in my previous review. The new draft has improved on an already excellent paper, and I look forward to seeing this published soon, as a valuable contribution to the intertemporal choice literature.

Final Decision Letter:

Dear Kai,

We are pleased to inform you that your Article "The globalizability of temporal discounting", has now been accepted for publication in *Nature Human Behaviour*.

Please note that *Nature Human Behaviour* is a Transformative Journal (TJ). Authors whose manuscript was submitted on or after January 1st, 2021, may publish their research with us through the traditional subscription access route or make their paper immediately open access through payment of an article-processing charge (APC). Authors will not be required to make a final decision about access to their article until it has been accepted. IMPORTANT NOTE: Articles submitted before January 1st, 2021, are not eligible for Open Access publication. Find out more about Transformative Journals

Acceptance of your manuscript is conditional on all authors' agreement with our publication policies (see <http://www.nature.com/nathumbehav/info/gta>). In particular your manuscript must

not be published elsewhere and there must be no announcement of the work to any media outlet until the publication date (the day on which it is uploaded onto our web site).

With best regards,
Aisha

Aisha Bradshaw, PhD
Senior Editor
Nature Human Behaviour